# GOFA: A Generative One-For-All Model for Joint Graph Language Modeling

**Lecheng Kong**[1*]  **Jiarui Feng**[1*]  **Hao Liu**[1*]  **Chengsong Huang**[1]  **Jiaxin Huang**[1]
**Yixin Chen**[1]  **Muhan Zhang**[2†]
[1]Washington University in St. Louis   [2]Peking University
`{jerry.kong, feng.jiarui, liuhao, chengsong, jiaxinh}@wustl.edu`
`ychen25@wustl.edu, muhan@pku.edu.cn`

## ABSTRACT

Foundation models, such as Large Language Models (LLMs) or Large Vision Models (LVMs), have emerged as one of the most powerful tools in the respective fields. However, unlike text and image data, graph data do not have a definitive structure, posing great challenges to developing a Graph Foundation Model (GFM). For example, current attempts at designing general graph models either transform graph data into a language format for LLM-based prediction or still train a GNN model with LLM as an assistant. The former can handle unlimited tasks, while the latter captures graph structure much better—yet, no existing work can achieve both simultaneously. In this paper, we first identify three key desirable properties of a GFM: self-supervised pretraining, fluidity in tasks, and graph awareness. To account for these properties, we extend the conventional language modeling to the graph domain and propose a novel generative graph language model GOFA. The model interleaves randomly initialized GNN layers into a frozen pre-trained LLM so that the semantic and structural modeling abilities are organically combined. GOFA is pre-trained on newly proposed graph-level next-word prediction, question-answering, structural understanding, and information retrieval tasks to obtain the above GFM properties. The pre-trained model is further instruction fine-tuned to obtain the task-solving ability. Our GOFA model is evaluated on various downstream datasets *unseen* during the pre-training and fine-tuning phases, demonstrating a strong ability to solve structural and contextual problems in *zero-shot* scenarios. The code is available at `https://github.com/JiaruiFeng/GOFA`.

## 1 INTRODUCTION

With the emergence of Large Language Models (LLMs), the field of artificial intelligence is undergoing a profound transformation, shifting from specialized, fragmented models to universal foundation models. A foundation model is pre-trained on large-scale datasets and can be further adapted to diverse downstream tasks using fine-tuning (Hu et al., 2022) or in-context learning (Bommasani et al., 2021; Touvron et al., 2023). Foundation models have been developed in different domains to handle text (Brown et al., 2020; Touvron et al., 2023), image (Kirillov et al., 2023; Bai et al., 2023), and even multi-modal data (Zhang et al., 2023c; Li et al., 2023; Alayrac et al., 2022). Because of their versatility and generalizability, foundation models have become prevalent in these domains.

However, despite preliminary efforts, a foundation model in the *graph domain* has arguably yet to be proposed. In the graph domain, data are highly flexible and dynamic. For example, social networks receive millions of new connections daily (Hardiman & Katzir, 2013), and novel molecules and protein structures are frequently discovered (Abramson et al., 2024; Gilmer et al., 2017). While past researchers have proposed specialized models to learn graph data (Ying et al., 2021; Kipf & Welling, 2017), the models require retraining to accommodate new graphs (Dai et al., 2022; Mo et al., 2022). Moreover, trained models are usually tied to specific applications and cannot be generalized to new domains and tasks. It becomes increasingly difficult for models to adjust to the ever-evolving

---

[*]Contributed equally. Listing order is random.
[†]Corresponding author

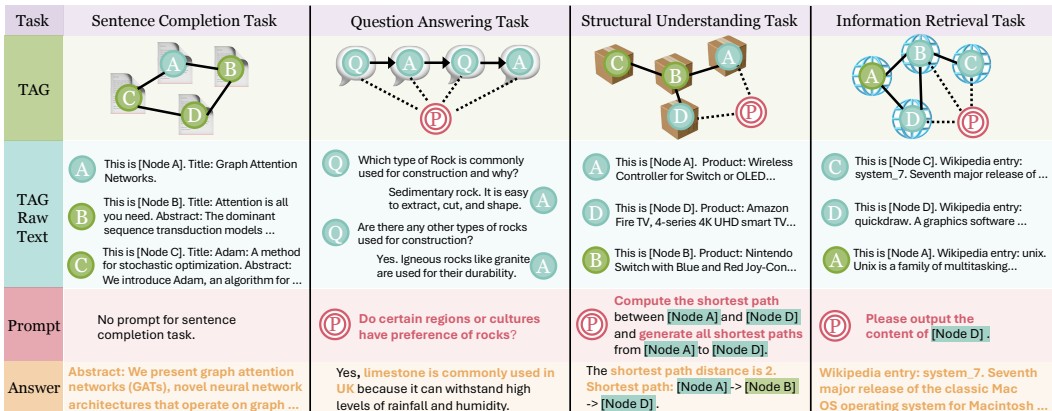

Figure 1: Examples of our pre-training tasks.

nature of graph data. Hence, a graph foundation model (GFM) applicable to new domains/tasks with minimal or no adaptation costs is urgently needed, spurring recent endeavors to study general graph models. In particular, a strong *zero-shot* ability is both challenging and fascinating for GFM researchers.

The success of LLMs inspired a series of preliminary attempts which use LLMs to develop general graph models. They can be roughly divided into two categories: LLM as a predictor and LLM as an enhancer (Chen et al., 2023). The **LLM as a predictor** approach transforms graph data into representations that LLMs can understand and use LLMs to generate predictions (Tang et al., 2023). However, as suggested by a recent study (Wang et al., 2023), *such an approach falls short of understanding graph structures*. This inspired the **LLM as an enhancer** approach, which adopts LLM to process and unify diverse graph data and feeds them to a GNN to train general graph models (Liu et al., 2023a; Huang et al., 2023a). Nevertheless, because GNN outputs fixed-sized representations/predictions, they can only handle specific tasks such as classification, and *cannot generalize to arbitrary, new tasks due to the lack of generation ability*. In summary, the current two approaches cannot fully utilize structural information and be generative simultaneously. We discuss the pros and cons of existing approaches in detail in Section 2.

In this paper, we first identify three desirable properties of a graph foundation model (GFM), namely large-scale self-supervised pre-training, fluidity in tasks, and graph understanding. To achieve the first property, we propose a generic graph self-supervised learning problem similar to the next-token prediction problem in LLMs, allowing label-agnostic and continual training on highly diverse graph data. We then propose a generative model termed Generative One-For-All (**GOFA**) that interleaves GNN layers into an LLM to achieve the second and third properties. Such a novel design systematically integrates GNN into an LLM, granting the LLM *graph structural learning ability* while keeping LLM's original free-form text generation ability. Meanwhile, this design allows the pipeline of the original LLM to remain intact, giving GOFA a *close-to-LLM level of task fluidity*. We pre-train the model with large-scale real-world graph data, Question-Answer (QA) chain data adopted from the NLP domain, and graph structural data to empower the model with the aforementioned foundational abilities in the graph domain (Examples in Figure 1). After pre-training, we further instruction fine-tune the model on a small amount of data (relative to the pre-training data) to make it understand task formats. The fine-tuned model is finally evaluated on various downstream datasets *unseen* during pre-training and fine-tuning. GOFA achieved impressive results on the zero-shot scenario, which demonstrates the strong potential of GOFA to serve as a graph foundation model.

## 2 A DESIRED FOUNDATION MODEL FOR GRAPH

In this section, we elaborate on three crucial properties a true graph foundation model should possess to motivate our GOFA model design. We note that many contemporary works (partly) propose similar ideas to ours and thus we do not claim the credit. We kindly refer readers to the latest surveys (Liu et al., 2023b; Jin et al., 2023; Zhang et al., 2023d) for more discussions on GFMs.

**Large-Scale Self-Supervised Pre-training:** One fundamental design of LLM is that it unifies all NLP tasks into a single next-token-prediction paradigm, which enables self-supervised pre-training on a large corpus collected from different sources. For pre-training graph models, while numerous efforts have been made from both the LLM as a predictor and LLM as an enhancer approaches, these attempts usually require the learning target to be labeled (Liu et al., 2023a; Chen et al., 2023). However, *a graph foundation model should have no constraint on the input graph (has labels or not) and can learn cross-domain knowledge from large-scale graph data in a self-supervised fashion.*

**Fluidity in Tasks:** A graph foundation model should also possess the same level of versatility and fluidity in handling different tasks as an LLM. Specifically, such ability can be broken down into three levels: (**a**) The graph foundation model can naturally respond appropriately to different graph tasks based on user instructions without requiring task-specific adjustment (e.g., the same model performs classification and question-answering tasks without any modification.) (**b**) With appropriate instruction-tuning, the model should have in-context learning ability on unseen tasks (e.g., a model tuned on citation network also performs well on knowledge graphs with proper instructions). (**c**) Users should be able to define new, previously unseen tasks by modifying the graph structure and features in a way that aligns with the universal input representation of the model. They can continuously train the model on new data without special adaptation. Existing approaches that use GNN models as the predictors are usually either restricted in the output format (Liu et al., 2023a; Xia et al., 2024; He et al., 2024a) or need additional fine-tuning on the task head (Sun et al., 2023; Wang et al., 2022). Consequently, despite having better structural modeling ability, such models cannot accommodate task changes or deal with novel tasks, e.g., shifting from a classification task to a question-answering task that requires outputting all shortest paths between two nodes.

**Graph Understanding:** Since the LLM as a predictor approach uses a generative LLM to take text input and produce text output, it naturally has the fluidity to accept varied prompts to tackle different tasks. However, such an approach processes the structural information poorly (Wang et al., 2023), making the utility of these models limited on many graph tasks. More importantly, even though some recent variants can use auxiliary graph models (such as GNNs) to incorporate structural information (Tang et al., 2023; He & Hooi, 2024; Zhang et al., 2024), the graph models are *frozen* and not responsive to different prompts, and the output from the graph models may not be the most relevant to the input prompt. On the contrary, *a graph foundation model should account for the unique structural information of graphs such as node degrees, shortest paths, common neighbors, etc., and generate graph representations dependent on the input prompt.* It should not only have LLM's prompt learning capability but also learn graph structure and semantic information jointly.

## 3 METHOD

In this section, we first propose a generative modeling framework for graphs, serving as the graph counterpart of traditional language modeling. Next, we introduce a novel GNN-LLM architecture for the proposed graph generative modeling problem. Finally, we describe the unified pre-training tasks to train GOFA towards the proposed GFM properties.

### 3.1 GENERATIVE MODELING FOR GRAPH

**Unifed task formats.** A generative model usually takes existing contexts, such as user prompts and passages, as input to generate conditional output related to the contexts, such as answers and completed sentences. Defining unified input and output formats for tasks in language applications is easy, as they are purely text-based. Further, because both the pre-training and downstream tasks are constructed in the same format (i.e., next-token-prediction), the downstream tasks conveniently adapt the knowledge from pre-training tasks, resulting in surprising capabilities, such as zero-shot learning. However, graph data from different domains vary significantly by input feature (e.g., nodes in a citation network have completely different vector representations as nodes in a knowledge graph) and output target, preventing direct knowledge transfer between tasks. Hence, the first challenge is to *define a unified format for graph tasks*, such that the model can do large-scale self-supervised pre-training on arbitrary graphs and transfer to downstream tasks seamlessly.

To **unify graph task input**, we follow the previous work OFA (Liu et al., 2023a) and extend the definition of Text-Attribute Graph (TAG) beyond graphs with text features such as citation and

product networks. In fact, any node and edge features can be represented by texts. For example, in airline networks, airport and flight route details can be converted into textual descriptions for nodes and edges. Non-textural features, like numerical data, can also be transformed into text strings, as in LLMs. Even for graphs without any features, we can still attach sentences like *"The degree of this node is 3"* to nodes. Formally, a TAG is a graph $G = \{V, E, X_V, X_E\}$ where $V$ and $E$ are the sets of nodes and edges. Each node $v \in V$ (edge $e \in E$) corresponds to a text description $x(v) \in X_V$ ($x(e) \in X_E$). Such a format encodes almost all existing graph data and serves well as a general input representation.

For self-supervised language modeling, the generated output essentially completes the input sentence. Such a task requires the model to have a deep semantic and logical understanding of the provided contexts, which is crucial for downstream applications. Similarly, in graph modeling, we aim to achieve the same level of understanding through graph completion tasks. Given a TAG, the output should complete the graph conditioned on its semantic and structural information. We choose to use natural language as the most tangible output format to complete a TAG. *Succinctly, all natural language tasks can be modeled as sentence completion, and similarly, we aim to model all graph tasks with graph completion.*

**Generative Graph Modeling.** We then formally define the generative graph modeling framework for graph completion. This framework supports various graph-related tasks, including classification and free-form question answering. An LLM starts generating only from the end of the input sentence. However, in a TAG, every end of a sentence on a node is a potential generation starting point, but users might only be interested in generating output for specific nodes. To accommodate this, we introduce Nodes of Generation (NOG), allowing users to specify starting points for generation. The modeling task is to take a TAG as input and complete the TAG logically and sensibly by completing the sentences on the potentially user-specified nodes.

We define *graph generative modeling* as the likelihood of the text $y$ associated with the NOG $v$:

$$p(y|v, G) = \prod_{l=1}^{L} p(y_l|y_{<l}, v, G), \tag{1}$$

where $y_l$ is the $l$-th token of $y$, and $y_{<l}$ is its preceding tokens. The NOG $v$ is a completion target node with initial corresponding text $x(v)$, and $x(v)$ can be empty. $G$ contains structural and textual information of neighbor nodes to help the model generate $y$. Under this framework, we can design a range of self-supervised learning tasks. For example, the graph completion task is shown on Figure 2, where the text on the NOG $v$ is incomplete, and the goal is to complete the sentence on it using the existing text and the neighbor information. This task is covered by Equation (1), which encourages the model to have a strong graph structure and feature comprehension ability. Thus, the importance of the framework is that a model properly solves such modeling problems can possess the three properties of GFM discussed in Section 2, thus can benefit diverse downstream tasks, even in the zero-shot fashion. Section F.2 discusses how the proposed framework applies to various tasks related to the three properties.

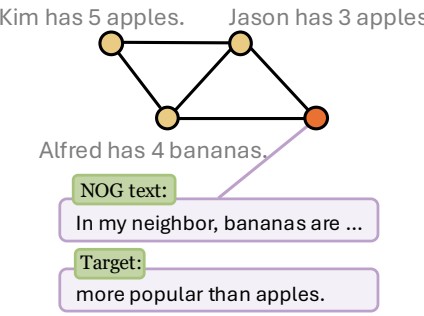

Figure 2: Task examples in TAG. Sentence completion/Next-word prediction. Orange node $v$ represents NOG.

## 3.2 GOFA : GENERATIVE ONE-FOR-ALL MODEL

To solve the generative graph modeling problem proposed in Equation (1), we design the GOFA architecture shown in Figure 3. Overall, GOFA consists of a *graph language encoder* and an *LLM decoder*. The graph language encoder interleaves GNN layers with LLM compressor layers to learn node representations containing joint structural and semantic information. The LLM decoder is then used to generate texts from the NOG representation. The LLM compressor and decoders are all pre-trained decoder-only transformers. We describe each component in detail as follows.

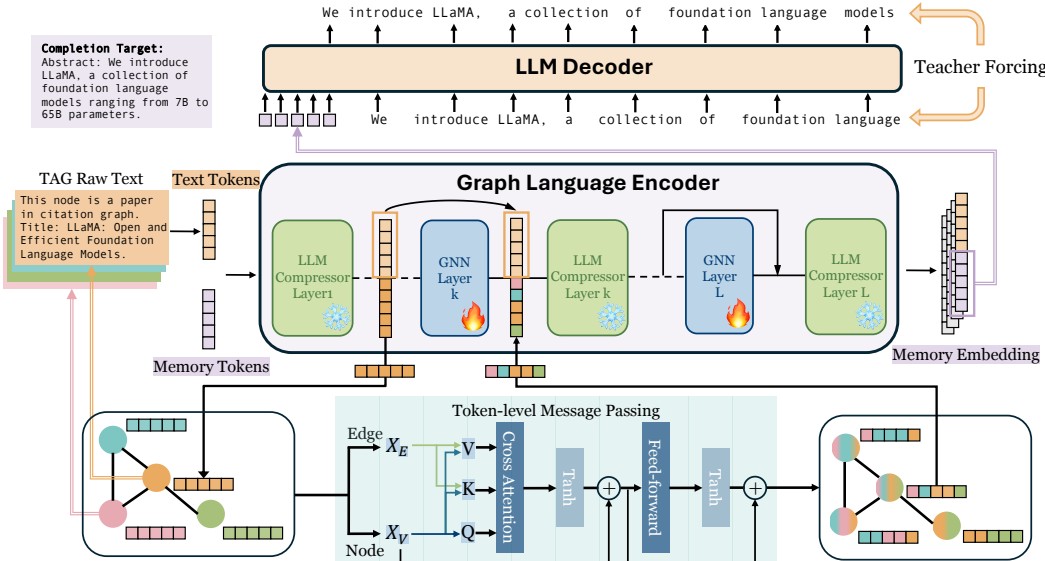

Figure 3: **GOFA Architecture**. Text tokens of TAG's node/edges are concatenated with memory tokens to be input to Graph Language Encoder. GNN layers are interleaved into LLM Compressor layers, where memory embeddings from LLM Compressor Layer are used as node/edge features for token-level GNN message passing. Memory embedding will be used for teacher-forcing training.

**LLM compressor:** Because GNNs require node and edge representations to have the same input dimension, many previous works propose to pool all tokens' output embeddings from the LLM as the node and edge vector representations and feed them to a GNN (Liu et al., 2023a; Huang et al., 2023a; He & Hooi, 2024). While this approach shows effectiveness in tasks of fixed form, such as classification and regression, *it is insufficient in more complex tasks such as generation*, as 1) the pooling process inevitably loses semantic information, 2) standard LLMs are not trained in a way such that the pooled output embedding is a summarization of the input sentence, and 3) the pooled representation space is no longer compatible with the space of the downstream LLM decoder. Hence, we adopt a pre-trained sentence compressor (Ge et al., 2023) that preserves as much information as possible from the original sentence in *fixed-size multi-token embeddings*. The core idea is to compress a sentence into $K$ embeddings instead of one embedding. Specifically, the sentence compressor has the same architecture as a decoder-only LLM, but the sentence to be compressed $\{q(x_i)\}_{i=1}^l$ is appended by a sequence of $K$ *memory tokens* $\{q(m_j)\}_{j=1}^K$, and the $t$-th layer of the LLM is:

$$\{Q_x^{t+1}, Q_{m,x}^{t+1}\} = \{q^{t+1}(x_1), ..., q^{t+1}(x_l), q^{t+1}(m_1), ..., q^{t+1}(m_K)\}$$
$$= LLM^t(\{q^t(x_1), ..., q^t(x_l), h^t(m_1), ..., h^t(m_K)\}) = LLM^t(\{Q_x^t, H_x^t\}). \tag{2}$$

We use $Q_x^t$ and $Q_{m,x}^t$ to represent the $t$-th LLM layer outputs corresponding to actual text tokens in sentence $x$ and the $K$ memory tokens appended at the end of text tokens, respectively. We use $H_x^t$ to represent the $t$-th GNN layer output, which will be explained later. In Equation (2), the text tokens ($Q_x^t$) and memory tokens ($H_x^t$, processed by the previous GNN layer) are concatenated as a single sequence of embeddings, which are fed to the current LLM layer. *Because the last $K$ tokens attend to all previous tokens, they can compress all information in the sentence into the output embeddings of the $K$ tokens.* This compressor architecture is inspired by ICAE (Ge et al., 2023). The compression ability is obtained through auto-encoder-style fine-tuning, as discussed in Appendix A.1.

**Token-level GNN:** Conventional GNNs take one embedding vector for each node/edge. However, now each node/edge sentence is compressed into $K$ memory token embeddings $Q_{m,x}$. Hence, we propose a simple extension of GNNs to the token level. For node $v \in V$, the $t$-th GNN layer is

$$H_{x(v)}^t[k] = GNN(Q_{m,x(v)}^t[k], \{(Q_{m,x(u)}^t[k], Q_{m,x(e_{uv})}^t[k])|u \in \mathcal{N}(v)\}), \quad k = 1...K. \tag{3}$$

In the GNN layer, tokens at different indices do not communicate. If we directly stack these GNN layers, they degenerate into multiple isolated GNNs for each token. Nevertheless, because we

interleave the GNN layers into the LLM layers, as shown in Figure 3, the isolated tokens exchange information in the subsequent self-attention layers of the LLM. This approach significantly reduces memory usage because we do not allow cross-token attention between different nodes. While edge memory tokens $Q_{m,x(e)}^t$ are passed into GNN to assist message passing, their representations are not updated in the GNN layer but directly passed to the next LLM layer, hence $H_{x(e)}^t = Q_{m,x(e)}^t$. In GOFA, we use a modified Transformer Convolutional GNN (Shi et al., 2021) to be consistent with the transformer architecture of LLM (see Appendix A.3 for details).

We insert one GNN layer between two transformer layers, while the first and the last layer are always transformer layers. In GOFA, we only insert GNN between the last few transformer layers, but this can be flexible depending on the computational resources. Following previous practice, we incorporate feed-forward (FF) layers into the GNN to increase expressivity and residual connections to stabilize training. Moreover, GOFA should maintain the functions of an LLM on plain texts, hence, inspired by the gating mechanism in earlier works (Hochreiter & Schmidhuber, 1997; Alayrac et al., 2022), we apply a $tanh$ gate, initialized at 0, to the GNN and FF layer outputs so that the *initial model ignores the information from GNN layers and is equivalent to the pre-trained LLM*. We introduce weight decay in the gating module to promote gate value staying in large non-zero values only when graph information helps generate more accurate final text outputs.

**LLM decoder:** After applying the model to the textual graph, the memory tokens $Q_{m,x}$ of every node *contain information about the text on the node, the surrounding node text, and the graph structure due to the message-passing process in the GNN layers*. Then, for the NOG $v$ and its corresponding target text $y$, we insert $Q_{m,x}$ at the front of the token embeddings of the target text to generate and use teacher-forcing to maximize the standard log-likelihood of $y$ using the next-token-prediction objective. In this way, we have modeled the problem in Equation (1). The compressor, decoder, and GNN parameters can be jointly or separately optimized, potentially with PEFT methods like LoRA (Hu et al., 2022). In this paper, we use ICAE (Ge et al., 2023) as our backbone LLM, but the GOFA architecture is not tied to any specific LLM. More details are discussed in Appendix A.2.

**Discussion.** Our proposed graph language encoder has several advantages over existing methods. Suppose a graph has $V$ nodes, $E$ edges, and the average number of tokens for all nodes is $k$. The complexity of one GOFA layer is $O(Vk^2)$, as the self-attention only happens within each node. Note that we have omitted the extra computation complexity of message-passing because it only happens at individual indices with $O(E) \ll O(Vk^2)$ in practical graphs. Instead, if we concatenate texts in all nodes and input them to a regular LLM, the complexity of one layer is $O((Vk)^2)$, which is significantly larger than GOFA. Further, introducing GNN layers in LLMs is theoretically more powerful than pure LLMs for modeling graph structures, which is discussed in Appendix E.2.

### 3.3 UNIFIED TASK REPRESENTATION IN GOFA

The formulation in Equation (1) provides a natural way for users to query the graph by selecting a NOG. Following OFA (Liu et al., 2023a), we convert all tasks into tasks on $k$-hop rooted subgraphs extracted around the target nodes. For node-level tasks, the target node is a single node in the graph. For link-level tasks, the target nodes are the node pair. If the target node is not specified (e.g., the task is a graph task), we set the default target nodes to all nodes in the graph. We connect a prompt node with the user query as NOG to all target nodes. GOFA completes the prompted input TAG by answering the query on the NOG, which still aligns with the proposed generative modeling framework. This design has several advantages: (1) All tasks are represented by a NOG, so the distribution of all tasks can be unified into a single space, helping the model generalize to unseen tasks from learned task representations; (2) The text feature for the prompt node describes the task details. Connecting the prompt node to target nodes enables the prompt node to query the most important knowledge from the input graph through attention. This ensures the output embedding for NOG is *conditionally learned from the GNN process subject to the different prompts*. Conversely, most of the previous works (He & Hooi, 2024; Tang et al., 2023; 2024; Zhang et al., 2024) only computed a fixed embedding for each node before any prompt is introduced.

### 3.4 LARGE-SCALE PRE-TRAINING

As discussed in Section 2 and Section 3.1, we design self-supervised pre-training tasks based on the three GFM properties to train GOFA. The training datasets include MAG240M (Hu et al., 2021a),

Pubmed and Arxiv (Hu et al., 2021b) for academic knowledge, Wikikg90mv2 (Hu et al., 2021a) and WikiGraph (proposed by us) for semantic diversity, and Ultrachat200k (Ding et al., 2023) dataset for question-answering ability. Details about the datasets can be found in Appendix C. Each node is assigned a unique ID (e.g., [Node A]) to enable node querying in the graph. We design four pre-training tasks as shown in Figure 1. We describe the rationale of each task below and leave some implementation details and additional discussion in Appendix F and Appendix E.3.

**Sentence Completion Task**. This task aims for large-scale pre-training (GFM property one) by training GOFA to predict the remaining text in a node based on both the existing node text and the surrounding graph information. Such a task can be applied to any TAG without labeling, thus facilitating large-scale pre-training for GOFA to acquire diverse knowledge.

**Structural Understanding Task**. This task aims to provide structural modeling ability for GOFA (GFM property three). The structural task connects NOG randomly selected node pairs to generate the actual shortest path or common neighbors between them. Through these two tasks, the model is expected to gain the ability to identify basic graph structures fundamental for graph-related problems.

**Question Answering Task**. This task aims to ensure fluidity in generation for GOFA (GFM property two). Unlike language corpus, which naturally contains many question-and-answer (QA) pairs, graph data usually only contain objective descriptions of entities. Hence, we convert natural language Question-Answer sequences into chain graphs and connect a NOG with a question to the chain graph for open-ended answer generation. This essential task enables GOFA to be responsive to arbitrary downstream applications expressed in free-form text questions.

**Information Retrieval Task**. In most downstream tasks, GOFA links a prompt node to target nodes in the graph to address related problems. To facilitate effective information extraction, we design an information retrieval task where a NOG queries a target node using its node ID. The model must retrieve and isolate information specific to the queried node from the remaining target nodes, encouraging a message-passing process conditioned on the input, as discussed in Section 3.2.

## 4 RELATED WORK

Here we mainly discuss the two tracks of general graph models, and leave discussion about graph prompt learning and graph neural networks to Appendix D.

**LLMs as enhancers:** One direction uses LLMs to convert the text features of graphs to unified representations (Liu et al., 2023a; Chen et al., 2023; Li et al., 2024; He et al., 2024a; Plenz & Frank, 2024) for downstream graph models to distinguish and transfer knowledge between different domains. For example, OFA (Liu et al., 2023a) uses LLM to unify the input features in different datasets and transforms multiple types of graph classification tasks into a unified binary classification format. TAPE (He et al., 2024a) utilizes LLM to generate question answers and explanations as enhanced node features. Such approaches have good structural modeling ability, but they usually cannot generate free-form output to handle arbitrary tasks.

**LLMs as predictors:** Another line of research proposes using LLMs as predictors and aligning graph representation with LLM inputs. Preliminary attempts flatten graphs into text representations and feed them into LLM (Chen et al., 2023; Zhao et al., 2023b; Guo et al., 2023; Zhao et al., 2023a; Qian et al., 2023). These approaches can benefit from LLM for task fluidity but fail to model structural information unique to graph data properly (Zhao et al., 2023b; Mao et al., 2024; Ye et al., 2023). Realizing this problem, follow-up work extends methods in vision-language domain (Alayrac et al., 2022; Li et al., 2023) to the graph domain and train adapters to link graph model outputs to LLM (Tang et al., 2023; 2024; Huang et al., 2024; Zhang et al., 2024; He & Hooi, 2024). For example, GraphGPT (Tang et al., 2023) first implements a text-structure alignment between graph representation and text embedding to pretrain a GNN. LLaGA (Chen et al., 2024) creatively uses a template to represent a subgraph with pooled node embeddings for LLM input. Inspired by Q-former (Li et al., 2023), GraphTranslator (Zhang et al., 2024) aligns node and text tokens from pre-trained GNN and LLM. UniGraph (He & Hooi, 2024) pretrains GNN using masked word prediction and then tuning a projector to map graph embedding to language space and enable zero-shot learning. However, the GNN and LLM parts of these methods are usually detached, meaning the prompt information can not attend to the message-passing process.

Table 2: Zero-shot experiment results with instruction tuning (Accuracy).

| Task | Cora-Node | | WikiCS | | Products | | | ExplaGraphs | Cora-Link |
|------|------|------|------|------|------|------|------|------|------|
| Way | 7 | 2 | 10 | 5 | 47 | 10 | 5 | 2 | 2 |
| LLama2-7B | 47.92 | 73.45 | 40.10 | 58.77 | 27.65 | 58.71 | 64.33 | 57.76 | 48.15 |
| Mistral-7B | 60.54 | 88.39 | 63.63 | 71.90 | 43.99 | 70.16 | 74.94 | 68.77 | 49.43 |
| OFA-Llama2 | 28.65 | 56.92 | 21.20 | 35.15 | 19.37 | 30.43 | 39.31 | 51.36 | 52.22 |
| GraphGPT | 44.65 | - | - | - | 18.84 | - | - | - | 50.74 |
| UniGraph | 69.53 | **89.74** | 43.45 | 60.23 | 38.45 | 66.07 | 75.73 | - | - |
| ZeroG | 64.21 | 87.83 | 31.26 | 48.25 | 31.24 | 51.24 | 71.29 | - | - |
| LLaGA | 51.85 | 62.73 | - | - | 23.10 | 34.15 | 39.72 | - | **88.09** |
| **GOFA-T** | **70.81** | 85.73 | **71.17** | **80.93** | 54.60 | 79.33 | 87.13 | **79.49** | 85.10 |
| **GOFA-F** | 69.41 | 87.52 | 68.84 | 80.62 | **56.13** | **80.03** | **88.34** | 71.34 | 86.31 |

## 5 EXPERIMENT

This section evaluates the proposed methods by answering the following four questions: **Q1**: Are the pre-training tasks in GOFA effective for graph-language modeling and structure understanding? **Q2**: Does the pre-trained GOFA help with critical general graph model application, zero-shot learning? **Q3**: Is using GOFA more advantageous than LLMs in graph tasks? **Q4**: Does GOFA have the fluidity to handle open-ended graph-related tasks? Additionally, we also include supervised experiments in Appendix F.5.

### 5.1 GOFA PRE-TRAINING

To answer **Q1**, we pre-train the GOFA model using ICAE models on Mistral-7B (Jiang et al., 2023), optimizing the objective in Equation (1) using the proposed tasks. The training details can be found in Appendix F.3. After training, we evaluate the perplexity of both GOFA and base LLM on Cora, Product, and Wikics datasets (all three are not included in the pre-training). We report the perplexity in Table 1. Note that during pre-training, we only update the weight of the GNN layers, and GOFA 's lower perplexity shows that the structural

Table 1: Evaluation for pre-trained GOFA . (RMSE for SPD and CN)

| | Perplexity ↓ | SPD ↓ | CN ↓ |
|------|------|------|------|
| Mistral-7B | 30.12 | 1.254 | 1.035 |
| GOFA-SN | 26.20 | - | - |
| GOFA | 21.34 | 0.634 | 0.326 |

and semantic information in the node's neighbor can effectively help complete the sentence with more relevance than the original LLM. Further, to validate that training of GOFA will not affect the original LLMs' ability, we input GOFA with single node graphs without any connections (denoted as GOFA-SN) to evaluate the perplexity, as shown in Table 1. We can see that without connection information around the center node, generation on a single node graph remains comparable to LLM and even better due to the pre-training process, showing that GOFA training does not destroy the desirable property of a pre-trained LLM. Besides sentence completion, another important GOFA pre-training objective is the structure learning ability. We report the shortest path distance and common neighbor count prediction results in Table 1, compared with LLM models whose inputs are textualized graphs with descriptions of edge connections. The datasets we used were Cora and Product. We see a significant performance improvement of GOFA over base LLM, showing that a difficult graph task for LLM can be well solved by the GNN layers with better structure modeling ability.

### 5.2 ZERO-SHOT LEARNING WITH GOFA

To answer **Q2**, we performed zero-shot experiments on various graph tasks. Despite using QA-chain data in the pre-training stage, the graph data does not include knowledge about task formats like classification and does not output exact matches to the answers. Hence, we first instruction-tuned the pre-trained GOFA in Section 5.1 on a small amount of data. We report the zero-shot results of two GOFA instruction tuning settings named GOFA-T and GOFA-F, as shown in Table 2 and Table 3. GOFA-T includes node and link classification tasks from Arxiv and Pubmed and GOFA-F addtionally adds MAG240M and Wiki90mv2 datasets. The instruction-tuning details can be found in

Table 3: Zero-shot experiment results with instruction tuning on FB15K237 and Scene Graphs.

| Task Format | FB15K237 10-Way | SceneGraphs QA |
|---|---|---|
| Llama2-7B | 48.32 | 38.62 |
| Mistral-7B | 62.48 | **45.95** |
| GOFA-T | 73.59 | 34.06 |
| GOFA-F | **80.69** | 31.36 |

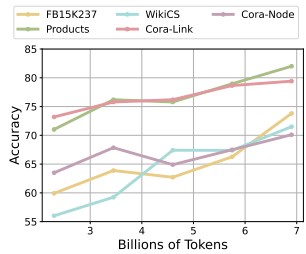

Figure 4: Performance vs pre-training sample size.

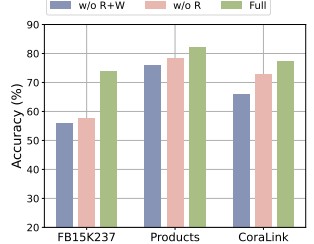

Figure 5: Pre-training Tasks Ablation Study.

Table 4: Comparison between GOFA and LLM with the same input.

| Task Metric | ExplaGraphs Acc ↑ | Time sec/sample ↓ | WikiCS Acc ↑ | Time sec/sample ↓ | Cora-Link Acc ↑ | Time sec/sample ↓ | FB15k237 Acc ↑ | Time sec/sample ↓ |
|---|---|---|---|---|---|---|---|---|
| LLM-N | 74.13 | 1.50 | OOM | OOM | 50.36 | 3.84 | 51.25 | 3.92 |
| GOFA-F | 79.49 | 0.48 | 71.17 | 2.43 | 85.10 | 1.67 | 73.49 | 3.37 |
| Improvement | 7.23% | 68.00% | NA | NA | 68.98% | 56.51% | 43.40% | 14.03% |

Appendix F.4. Note that the zero-shot datasets are unseen during both pre-training and instruction finetuning. The goal of instruction fine-tuning is not to let the model learn particular knowledge from these datasets but to make the model understand the task format described in Appendix F.4.

While the instruction-tuning dataset only covers the relatively small spectrum of the graph datasets, we observe that GOFA achieves very non-trivial performance on all node-level (Cora-Node, WikiCS, Products), link-level (FB15K237, Cora-Link), and graph-level (ExplaGraphs, SceneGraphs) tasks. GOFA also generalizes to different ways and even question-answering (SceneGraphs) tasks, showing its desirable fluidity. GOFA outperforms LLM and graph foundation model baselines on most datasets and exceeds best baselines by a large margin ($> 10\%$) on WikiCS, Products, FB15K237 and ExplaGraphs, showing GOFA's ability to combine the advantage of both LLM and graph models. GOFA not only achieves remarkable results on the knowledge graph and academic graph, which are proximal to the trained data but also excels in Products and ExplaGraphs whose distribution shifts significantly from training data, which further highlights GOFA 's substantial generalizability. Meanwhile, we observe that GOFA is only achieving comparable performance to LLM on the SceneGraph dataset. We suspect that the instruct-tuning data contains information-dense texts, reducing the model's ability on common sense questions that this dataset requires. In the future, we plan to diversify instruction-tuning datasets with common sense knowledge to enhance such ability.

We further conducted the same experiments on intermediate pre-training checkpoints, and show results in Figure 4. We observe that as the model witnesses more pre-training samples/tokens, the downstream task performance also increases significantly, confirming the importance of large-scale pre-training on graph data. The performance continues to improve, meaning that the model can potentially scale to higher capability with more samples; we leave this to future work. In Figure 5, we plot the instruction-tuning performance when we remove the Wikipedia datasets and information retrieval task (w/o R+W), only remove the retrieval task, (w/o R), and full tasks. We can see that Wikipedia datasets improve the model performance of all the datasets for the diverse corpus it introduced. The retrieval tasks particularly improve the knowledge graph performance due to the improved ability to retrieve key correlations between target entities. These show the necessity and effectiveness of the overall pre-training task selection and design. In Appendix B.1, we further conduct zero-shot experiments on molecule datasets.

## 5.3 COMPARING GOFA WITH LLMS

Answering **Q3** is critical to understanding the necessity of the GNN layers and the effectiveness of GOFA as a general graph model. We compare GOFA to LLM whose textual prompt contains the same information as the input graph to GOFA. Specifically, for a GOFA input graph, we concatenate all node texts as the prompt and append the connection information to it, as in *"Node A connects*

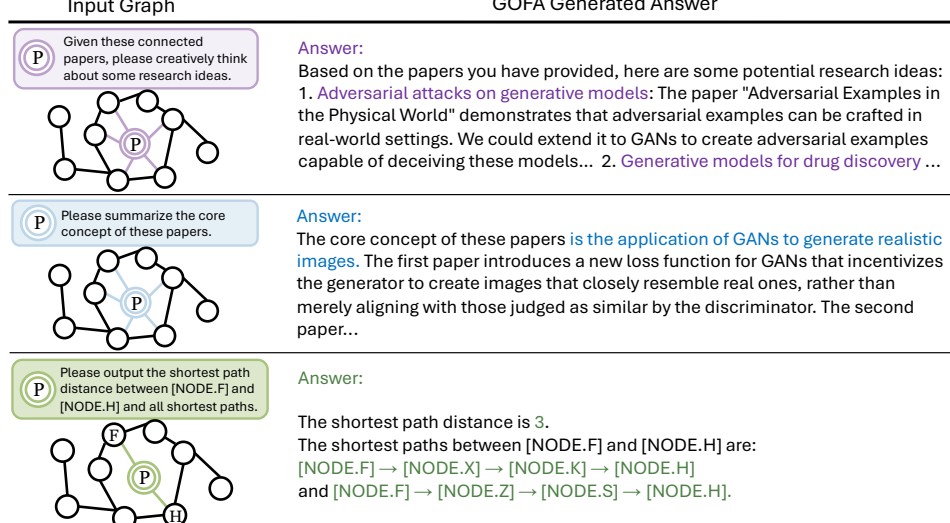

Figure 6: GOFA diverse responses to open-ended questions.

*to Node B".* The text is then combined with task and question descriptions as input to an LLM for classification tasks. Approaches similar to this are widely adopted and acknowledged (Chen et al., 2023; Fatemi et al.). We present both the classification performance and per sample inference time in Table 4 and denote the LLM method as LLM-N. We observe impressive performance improvement of GOFA on all datasets, even when the LLM is prompted with the same information, showing that GOFA, with the help of the GNN and interleaving design, utilizes the graph information much more effectively. Moreover, we also observe a fundamental reduction in inference costs, confirming our analysis in Section 3.2 that, with the same input, GOFA is more efficient than LLMs. Note that when the input size is large, such as in WikiCS, LLM struggles with high memory consumption of the long sequence, whereas the GOFA avoids that by leveraging the sparsity of graph data and using edge information to compute the most important attention information.

## 5.4 GOFA Responses on Diverse Tasks

Finally, we answer **Q4** by providing generation examples of GOFA in Figure 6, where we prompt the same citation graph differently and achieved corresponding and high-quality responses. The top and middle examples have the same connection for their NOGs (both connected to the same five nodes), but when we change the prompt text on the NOGs, the generated texts also adjust accordingly, utilizing the neighbor node information, validating that the message-passing is conditioned on the prompt. As in the bottom example, we can also prompt the graph differently by connecting the NOG to two target nodes and querying about the shortest path distance. In this case, the model successfully generates actual paths between the two nodes, which is an ability not seen in traditional graph models that can only output numerical predictions about the path length. These examples demonstrate GOFA's outstanding ability to answer open-ended questions. More examples are provided in B.4.

## 6 Conclusion, Limitations, and Future Works

We introduce GOFA, a generative One-for-All graph foundation model. GOFA is pre-trained under a graph completion framework to enable large-scale self-supervised learning. By integrating GNN layers with LLM layers, GOFA combines the generative capabilities of LLMs for free-form output with the structural learning strengths of GNNs for understanding complex graph connections. Our experiments demonstrate that GOFA, when fine-tuned with a small number of data, achieves impressive zero-shot performance in unseen datasets. However, our zero-shot experiment mainly focuses on evaluating models on unseen datasets with similar task formats instead of unseen task formats. Additionally, we employ a frozen LLM compressor in GOFA ; hence, the compression capability is not naturally unified with the graph data, potentially impacting the adaptability of the model. We will explore harder zero-shot settings and better fine-tuning strategies in the future.

## 7 ACKNOWLEDGEMENT

This work is partially supported by the National Key R&D Program of China (2022ZD0160300) and NSF grant CBE-2225809.

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

# APPENDIX

## A  IMPLEMENTATION DETAILS

### A.1  IN-CONTEXT AUTOENCODER (ICAE)

This section briefly introduces ICAE and how it helps build the GOFA model; please refer to ICAE paper (Ge et al., 2023) for the specifics of the model. ICAE contains two decoder-only LLMs. One serves as a language compressor that compresses sentences into a fixed-length sequence of vectors, and the other serves as a language decoder that decodes or queries into the compressed sentence representations. Specifically, during training, an input token sequence $x = \{x_1, ..., x_l\}$ is appended by a $K$ memory tokens $\{m_1, ..., m_k\}$ with trainable embeddings. The concatenated sequence is fed to the LLM compressor with a LoRA adapter (Hu et al., 2022).

$$\{h(x_1), ..., h(x_l), h(m_1), ..., h(m_K)\} = LLM_{comp}(\{e(x_1), ..., e(x_l), e(m_1), ..., e(m_K)\}), \quad (4)$$

where $e(\cdot)$ and $h(\cdot)$ are the token embeddings and LLM outputs. Then, the decoder LLM only attends to the memory token outputs and tries to decode the original sentence from the memory tokens.

$$\{l(m_1), ..., l(m_K), l(x_1), ..., l(x_l)\} = LLM_{dec}(\{h(m_1), ..., h(m_K), e(x_1), ..., e(x_l)\}),$$
$$\min_{\Theta_{comp}} \text{CrossEntropy}(\{l(m_K), l(x_1), ..., l(x_{l-1})\}, \{x_1, ..., x_l\}). \quad (5)$$

The ICAE model is also trained on QA and Language modeling tasks to have more diverse embeddings.

By training this auto-encoder objective on a large-scale, the compressor model learns to compress all information about a sentence to the memory token outputs like in a conventional auto-encoder model. In Table 5, we provide a few examples of the comparison between the original text and the text decoded from the compressed memory tokens by ICAE's decoder. Because the compressed representation contains as much information as possible, GNN can pass messages between nodes with minimal information loss.

Table 5: Comparison between original texts and decoded text from the compressed representation.

| Original Text | Decoded Text |
| --- | --- |
| Actress Halle Berry has been sharing a number of stunning photos from the time she has spent in Morocco and she just posted a new one to her Instagram page that fans will not want to miss. | Halle Berry has been sharing a number of stunning photos from the time she has spent in Morocco and just posted a new one on her Instagram page that fans won't want to miss. |
| Utah avoided the turnover bug on Saturday for the first time since its season opener. In addition, the running game was clicking and the defense was dominant as the Utes snapped a two-game winning streak on the road, beating Pittsburgh 26-14. Five keys to Utah's victory: 1. Utah running back John White IV: Running strong and with purpose from the beginning, White was a big reason why the Utes were within striking distance at halftime. White, who took a couple pops that dislodged his helmet and caused a cut below his ear, seemed to get stronger as the game wore on. He finished the afternoon with 171 yards on 36 carries. | Utah avoided the turnover bug on Saturday for the first time since its season opener. In addition, the running game was clicking and the defense was dominant as the Utes snapped a two-game winning streak on the road, beating Pittsburgh 26-14. Five keys to Utah's victory: 1. 2. 3. 4. 5. Utah running back John John White IV IV ran strong and with purpose from the beginning, being a big reason why the Utes were within striking distance at halftime. He took a couple of shots that dislodged his helmet and caused a cut below his ear, but seemed to get stronger as the game went on. He finished the afternoon with 171 yards on 36 carries. |

### A.2  LLM CHOICES OF GOFA

Because ICAE preserves as much information in a sentence as possible, we can use it in the GOFA model to comprehensively pass information between sentences, as shown in Section 3.2. However,

the GOFA model is not limited to ICAE. Users can first train an ICAE-like objective on any existing LLM and apply the GOFA model to the trained LLM. Or, users can apply the GOFA directly to a pre-trained LLM and train the GOFA without the auto-encoder training. Note that the ICAE architecture has a function similar to an encoder-decoder LLM. We do not use an off-the-shelve encoder-decoder LLM because its encoder output is still subject to the sentence length, which does not fit GNN's need for fixed-sized input.

The design of GOFA can be extended beyond a compressor-decoder architecture. For example, we can have a decoder-only GOFA whose LLM layer is,

$$\{Q_x^{t+1}, Q_{m,x}^{t+1}, Q_y^{t+1}\} = LLM^t(\{Q_x^t, H_x^t, Q_y^t\}), \tag{6}$$

where the GNN is still applied on $K$ memory tokens inserted between the node text $x$ and target text $y$. This allows the target text to attend to the node text, which may improve the performance of GOFA. However, this formulation forces every node to have a target text, which is usually not what users desire and poses extra computation costs. We will explore this architecture in our future work.

### A.3 TRANSFORMER CONVOLUTIONAL GNN

As mentioned in Section 3.2, we customize a Transformer Convolutional GNN(TransConv) (Shi et al., 2021) as the GNN used in Equation 3. Since GNN layers operate on token representations and tokens at different indices do not communicate, we describe the GNN at one index for simplicity. The $t$-th GNN layer on node $i$ and its neighbors $\mathcal{N}(i)$ is:

$$h^{t+1}(i) = \boldsymbol{W}_o(\sum_{j \in \mathcal{N}(i)} \alpha_{i,j}(\boldsymbol{W}_{v,node}h^t(j) + \boldsymbol{W}_{v,edge}h(e_{i,j}))),$$

$$\alpha_{i,j} = \text{Softmax}(\frac{\boldsymbol{W}_q h^t(i) * (\boldsymbol{W}_{k,node}h^t(j) + \boldsymbol{W}_{k,edge}h(e_{i,j}))}{\sqrt{d}}), \tag{7}$$

$h(\cdot)$ represents input node and edge features. $\boldsymbol{W}$ represents query ($q$), key ($k$), value ($v$), output ($o$) linear projection for nodes and edges. The formulation closely follows the transformer design (Vaswani et al., 2017) and its GNN adaptation (Shi et al., 2021). This formulation does not aggregate the last layer embedding $h^t(i)$ into the next layer, because we already add residual to maintain the same effect. We use pre-layer normalization following Llama (Touvron et al., 2023).

## B    ADDITIONAL EXPERIMENTS

### B.1    ADDITIONAL ZERO-SHOT EVALUATION RESULTS

Table 6: Additional comparison on zero-shot learning results. (The number indicates the number of ways).

|  | Cora-node (7) | WikiCS (10) | Products (10) |
|---|---|---|---|
| DGI | 20.15 | 13.63 | 21.35 |
| GraphMAE2 | 37.08 | 19.03 | 25.46 |
| Best-LLM | 60.54 | 63.63 | 70.16 |
| Best Graph-LLM | 69.53 | 43.45 | 66.07 |
| GOFA-T | **70.81** | **71.17** | **79.33** |

Table 7: Zero-shot learning with molecule dataset.

|  | BBBP | HIV |
|---|---|---|
| OFA | - | 35.67 |
| MoMu | 49.81 | 50.26 |
| Galactica | 53.94 | 33.85 |
| GIMLET | 59.39 | 66.24 |
| GOFA | 54.91 | 53.02 |

To further demonstrate the advantage of GOFA compared to previous methods, we conduct additional zero-shot experiments over different datasets and baselines. First, we additionally compare the zero-shot result with traditional GNN baselines. Since models like GCN and GAT cannot do zero-shot tasks due to their supervised training nature, we compare against graph self-supervised learning baselines. These methods learn node embeddings, which are compared to label text embeddings to make predictions based on cosine similarity. Specifically, we include DGI Veličković et al. (2018) and GraphMAE2 Hou et al. (2023) and follow the same evaluation procedure. The result is shown in Table 6. GOFA significantly outperforms methods that use GNNs for self-supervised learning in the zero-shot setting. This result further demonstrates the effectiveness of the architecture design and the graph modeling tasks of GOFA on zero-shot generalization ability over graph tasks.

Next, we evaluate the GOFA on more numerically and structurally intensive tasks. Specifically, we conduct experiments on molecular property prediction tasks using ogbg-molhiv and BBBP datasets. The experiments closely follow the setting of GIMLET Zhao et al. (2023a). First, we fine-tune the pre-trained GOFA on randomly sampled 100,000 question-answering pairs from the Chembl pretrain dataset. The questions concentrated on asking about different molecule properties. Next, we evaluate the fine-tuned GOFA on ogbg-molhiv and BBBP in a zero-shot setting. We use AUROC as the evaluation metric. For comparison, we include OFA Liu et al. (2023a), MoMu Su et al. (2022), GIMLET Zhao et al. (2023a), and Galactica Taylor et al. (2022). The results are shown in Table 7. We directly report results of OFA Liu et al. (2023a) from their paper, and all other baselines are referred from GIMLET Zhao et al. (2023a). We can see that with only 100,000 instruction-tuning samples, GOFA already achieves comparable results to baselines. Note that GIMLET fine-tunes its model on the whole Chembl-pretrain dataset, which contains more than 400 million question-answering pairs.

## B.2 SUPERVISED EXPERIMENT RESULTS

In this section, we conduct a supervised learning experiment with the pre-trained GOFA . In the supervised experiment, GOFA 's prompt does not include class optional. We show the supervised results in Table 8. Specifically, we compare the result of GOFA with the following baselines: 1. basic GNNs, which are trained individually on each dataset, including GCN (Kipf & Welling, 2017) and GAT (Veličković et al., 2018). 2. The contrastive learning methods, including DGI (Veličković et al., 2018) and BGRL (Thakoor et al., 2021). For these methods, we directly report the best result from (He & Hooi, 2024). 3. Graph foundation model, including OFA (Liu et al., 2023a) and UniGraph (He & Hooi, 2024). GOFA achieved competitive performance on most datasets. In particular, GOFA achieved SOTA performance on the Pubmed dataset, demonstrating that GOFA can transfer pre-trained knowledge to downstream tasks. We also notice that GOFA is not performing as well on some datasets, possibly because in a supervised setting, we only train a small portion of the data for one epoch (specific numbers in the experimental details section in Appendix F.5), and in the supervised setting, it is important to see training samples multiple times to ensure detailed understanding of the distribution. As we pre-train the model with more diverse datasets, GOFA can potentially obtain world knowledge as an LLM, which makes transfer learning in the supervised setting more accurate.

Table 8: Experiment results in supervised learning. **Bold** and underlined shows best and runner-up results.

| Task type
Metric | Cora
Link
Acc ↑ | Cora
Node
Acc ↑ | PubMed
Link
Acc ↑ | PubMed
Node
Acc ↑ | Arxiv
Node
Acc ↑ | WikiCS
Node
Acc ↑ | WN
Link
Acc ↑ | FB
Link
Acc ↑ | Products
Node
Acc ↑ |
|---|---|---|---|---|---|---|---|---|---|
| GCN | $78.9_{\pm0.6}$ | $\mathbf{82.3}_{\pm1.1}$ | $77.5_{\pm0.4}$ | $77.8_{\pm0.7}$ | $73.9_{\pm0.6}$ | $77.0_{\pm0.6}$ | $82.7_{\pm0.4}$ | $90.1_{\pm0.3}$ | $80.0_{\pm0.7}$ |
| GAT | $80.1_{\pm0.3}$ | $80.4_{\pm0.4}$ | $80.5_{\pm0.2}$ | $76.6_{\pm0.5}$ | $\mathbf{75.8}_{\pm0.3}$ | $79.8_{\pm0.5}$ | $88.8_{\pm0.3}$ | $93.6_{\pm0.1}$ | $\mathbf{81.4}_{\pm0.2}$ |
| DGI | - | $51.99_{\pm0.45}$ | - | $55.76_{\pm0.56}$ | $55.21_{\pm0.21}$ | $67.11_{\pm0.12}$ | $52.04_{\pm0.22}$ | $26.99_{\pm0.22}$ | $64.21_{\pm0.32}$ |
| BGRL | - | $56.73_{\pm0.23}$ | - | $63.77_{\pm0.23}$ | $62.21_{\pm0.21}$ | $70.12_{\pm0.15}$ | $56.44_{\pm0.21}$ | $64.91_{\pm0.22}$ | $63.77_{\pm0.23}$ |
| OFA | 87.97 | 75.34 | **95.89** | 77.89 | 73.44 | 77.62 | **98.31** | 95.78 | - |
| UniGraph | - | $81.43_{\pm0.55}$ | - | $74.33_{\pm0.23}$ | $72.91_{\pm0.42}$ | $\mathbf{79.98}_{\pm1.21}$ | $85.45_{\pm0.34}$ | $94.81_{\pm1.32}$ | $80.11_{\pm0.23}$ |
| GOFA | **89.54** | 76.50 | 93.97 | **83.83** | 74.77 | 79.96 | 92.16 | 88.21 | 79.98 |

## B.3 EVALUATION ON ADDITIONAL GRAPH STRUCTURAL TASKS

To further evaluate the potential of GOFA to serve as a general-purpose graph foundational model, we conducted additional experiments on a variety of graph structure tasks, including node degree, shortest path distance, and node count. To provide a fair comparison, we selected Mistral-7B as the baseline model and designed the prompts for Mistral following the incident prompt format proposed in (Fatemi

Table 9: Evaluation on additional structural tasks (Acc).

|  | Node degree | SPD | Node Count |
|---|---|---|---|
| Mistral | 98.6 | 99.8 | 99.9 |
| GOFA | 98.2 | 99.7 | 86.3 |

et al.). Specifically, the graph is described as "[NODE.A] is connected to [NODE.B]". We use Arxiv as the graph dataset and randomly selected 10000 samples for fine-tuning and 2000 samples for testing. For the mistral model, we remove all original text from the dataset, as the model will run out of memory with all the text kept. The evaluation result is shown in Table 9. We can see that

GOFA achieved a competitive performance to the LLM baseline in both node-level (node degree) and link-level (shortest path distance) structural tasks. However, LLMs perform better on graph-level tasks. We suspect that the reason lies in the fact that we keep the original text in the graph dataset, which makes the learning of structure much harder, especially when the number of nodes in the graph becomes large. At the same time, our pre-training tasks focus on understanding local graph structures. Thus, GOFA is unfamiliar with understanding global structural information like node count. In the future, we aim to include more structural tasks to further improve the generalization ability of GOFA on different graph tasks.

### B.4   EXAMPLE OF GOFA'S FREE-FORM ANSWER

Figure 6 in the main body illustrates GOFA's capability to respond to various questions based on the same graph from ogbn-arXiv. In this section, we provide additional examples in Figure 7 to further show the ability of GOFA's free-form text answer.

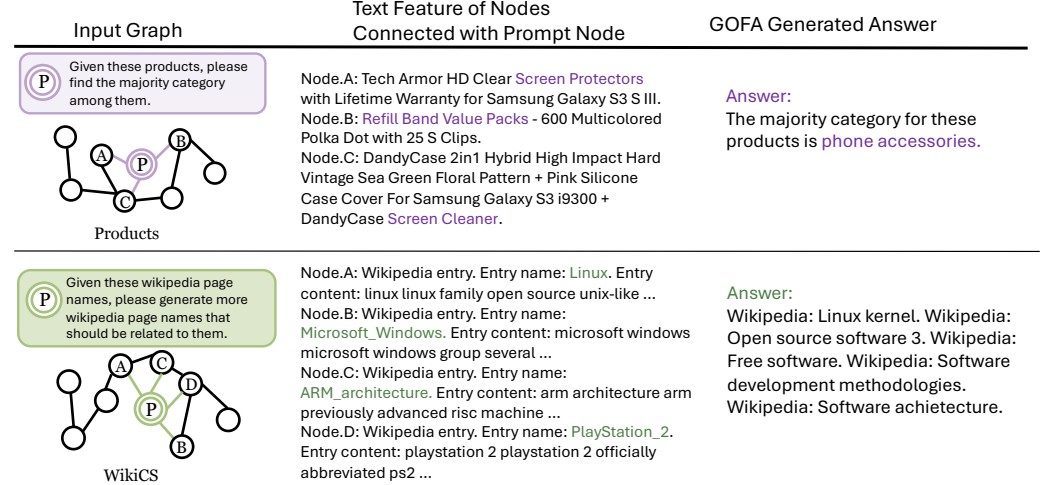

Figure 7: Demonstration of GOFA's ability to respond to any question to the given graph. Above is an example of the products dataset, where the model needs to output the majority category of its connected nodes. Below is another example on Wikics dataset. GOFA is asked to generate a Wikipedia page named based on the graph information.

## C   DATASETS

**Cora**. The Cora dataset is a co-citation network, where nodes are papers related to artificial intelligence. Edges mean the connected two papers are co-cited by other papers. The Cora dataset contains 2708 nodes and 10556 edges. We collect the Cora dataset and its raw text from OFA (Liu et al., 2023a). We evaluate the performance of the baseline and our proposed model on Cora for both node-level and link-level tasks. For the node-level task, the aim is to classify the node into the correct paper category from 7 different categories. The split is obtained from OFA. It contains 140/500/2068 samples for train/val/test set respectively. For the link-level task, the object is to predict whether two paper nodes are co-cited or not. We follow the setting of OFA (Liu et al., 2023a) and randomly split all edges into train/val/test sets with a ratio of 0.85/0.05/0.1.

**PubMed**. The PubMed dataset is a co-citation network, where nodes are papers related to diabetes mellitus. Edges mean the connected two papers are co-cited by other papers. The PubMed dataset contains 19717 nodes and 88648 edges. We collect the PubMed dataset and its raw text from OFA (Liu et al., 2023a). We evaluate the performance of the baseline and our proposed model on PubMed for both node-level and link-level tasks. For the node-level task, papers have 3 different categories. The goal is to classify the node into the correct paper category. We obtain the split directly from original source. It contains 60/500/19157 samples for train/val/test set respectively. For the link-level task, the object is to predict whether two paper nodes are co-cited or not. We follow the

setting of OFA (Liu et al., 2023a) and randomly split all edges into train/val/test sets with a ratio of 0.85/0.05/0.1.

**Arxiv**. The Arxiv dataset is a citation network, where nodes are papers related to computer science and edges mean two papers have a citation relationship. The Arxiv dataset contains 169343 nodes and 1166243 edges. We collect the Arxiv dataset and its raw text from OGB (Hu et al., 2021b). We evaluate the node classification on the Arxiv dataset. The goal is to classify the paper node into the correct category from 40 possible categories. We obtain the split directly from OGB (Hu et al., 2021b). It contains 90941/29799/48603 samples for train/val/test set, respectively.

**WikiCS**. The WikiCS dataset is a graph obtained from Wikipedia. The nodes in WikiCS are Wikipedia terms and their descriptions. The edges mean there is a hyperlink between two terms. We collect the WikiCS dataset and its raw text from (Mernyei & Cangea, 2020). There are 11701 nodes and 216123 edges in the graph. We evaluate the performance of WikiCS on the node classification task. There are 10 different classes. We follow the same split as OFA (Liu et al., 2023a), which contains 580/1769/5847 samples for the train/val/test set, respectively.

**Products**. The Products dataset is a co-purchase graph. The nodes in the graph represent product items from the Amazon platform, and the edges represent that two products are co-purchased together. We obtain the Products and their raw texts from TAPE (He et al., 2024a), which is a subset from the original ogbn-Products (Hu et al., 2021b) dataset. It contains 54025 nodes and 144638 edges. We evaluate the node classification performance on Products. The data from the original source contains 47 different categories. However, we found that there are two classes with missing labels. To be consistent with previous literature, we adopt the approach in LLaGA to replace the label name with special symbols. There are 14708/1572/37745 samples for the train/val/test set, respectively.

**FB15K237**. The FB15K237 is a knowledge graph generated from Free Base. Nodes in the dataset represent entities in the world and edges represent the relation between entities. We obtained the dataset from OFA (Liu et al., 2023a). The FB15K237 is used to evaluate the link classification. The dataset contains 237 unique classes. We follow the setting of OFA (Liu et al., 2023a) and split the dataset with a ratio of 0.85/0.05/0.1, which results in a total of 272115/17535/20466 samples for train/val/test set, respectively.

**ExplaGraphs**. The ExplaGraphs is a graph question answering dataset on commonsense concepts. Nodes in the dataset represent a common sense concept and edges represent the relation between two concepts. We obtain the dataset from G-retriever (He et al., 2024b) The ExplaGraphs can be used for question-answering on graphs. We obtain the split directly from G-retriever (He et al., 2024b). It contains 1659/553/554 graph samples from the train/val/test set.

**SceneGraphs**. The SceneGraphs is a graph question answering dataset on scene graphs. Nodes in the dataset represent an object in an image and edges represent the relation between two objects. We obtain the dataset from G-retriever (He et al., 2024b) The SceneGraphs can be used for question-answering on graphs. We obtain the split directly from G-retriever (He et al., 2024b). It contains 59978/19997/20025 graph samples from the train/val/test set.

**MAG240M**. The MAG240M dataset is a citation network generated from Microsoft Academic Graphs. The nodes represent academic papers and the links represent a citation relation between two papers. We obtained the dataset and raw text from OGB-lsc (Hu et al., 2021a). However, the original dataset is extremely large and contains nodes without text features (author and institution nodes), since we mainly use the dataset for pre-training, we further downsample the original dataset. Specifically, we only keep paper nodes and citation links between papers. Further, we downsample the edges in the following ways. First, we selected all nodes in the train/val/test split provided by OGB-lsc (Hu et al., 2021a). Next, we filter the edges through two rounds. In the first round, we only keep the edge if either the source or the target is in the selected nodes. If any node in the added edge is not in the selected nodes, we add it to the node set. Next, in the second round, we include additional edges where both the source and target are in the selected nodes (additional nodes are added in the first round). The procedure results in a total of 5875010 nodes and 26434726 edges.

**Ultrachat200k**. The Ultrachat200k is a question-answering dataset. each sample is a multi-round conversation obtained from the web. We obtained the Ultrachat200k from (Ding et al., 2023). However, the original dataset is not a network. To convert it to a graph dataset, we manually create a graph structure for it. Specifically, if the original sample has $k$ round of conversation, we will

generate $k - 1$ graph sample. The $i$-th graph will contain the first $i$ round of conversation. Each node in the graph is either a question or an answer. The question and answer are linked by a directed edge indicating the order of the conversation. The conversation of $i + 1$ round will be the question-answer pair for this graph. Since we mainly use the dataset for pre-training. We only include *train-sft* subset. After the conversion, there are a total of 449929 graphs in total.

**Wikikg90m**. Wikikg90m is an encyclopedic knowledge graph dataset extracted from Wikidata knowledge base. We obtain the original Wikikg90m from OGB-LSC (Hu et al., 2021a). It contains 91,230,610 entities, 1,387 relations, and 601,062,811 edges.

**WikiGraph**. The WikiGraph dataset is designed to increase the diversity of the training texts. Hence, we use WikiText (Merity et al., 2022) dataset as the seed dataset. It contains plain sentences from Wikipedia pages. We generate graphs for sentences with more than 500 characters. Specifically, we first prompt an LLM to extract meaningful entities or concepts from a sentence, and these entities become the nodes in the graph. We then randomly pair concepts to generate edges. Again, we use LLM to generate a description of the relationship between the paired concepts and use the description as the edge text.

**BBBP**: The BBBP is a molecule property prediction dataset. The task is to predict the effectiveness of molecule to brain blood barrier. We collect the dataset from GIMLET Zhao et al. (2023a). The dataset contains 1631/204/204 molecule graphs for the train/validation/test set, respectively. We directly use the test set for zero-shot evaluation.

**HIV**: The HIV dataset is a molecule property prediction dataset. The task is to predict the effectiveness of molecules to hiv. We collect the dataset from GIMLET Zhao et al. (2023a). The dataset contains 32901/4113/4113 molecule graphs for the train/validation/test set, respectively. We directly use the test set for zero-shot evaluation.

**Chembl pretrain**: The Chembl pretrain dataset is a large-scale molecule property prediction dataset. for each simple molecule, the dataset contains several questions to ask about the different properties of the molecule. Overall, it contains 400 million question-answering pairs. We collected it from GIMLET Zhao et al. (2023a) and use it for fine-tuning GOFA for molecule zero-shot experiments.

## D  RELATED WORK EXTENDED

**Prompt Learning in Graph:** The success of foundation models inspired many works to adapt their power to the graph domain. Earlier attempts designed a graph prompting mechanism such that a trained model can adapt to new data by fine-tuning a soft prompting vector (Liu et al., 2023c; Yu et al., 2023; Sun et al., 2023; Xia et al., 2024). GraphPrompt (Liu et al., 2023c; Yu et al., 2023) pretrains on link prediction tasks, and then finetune a prompt matrix for each downstream node or graph classification task. All in One (Sun et al., 2023) designs prompt tokens that are used to modify node features and then take a meta-learning paradigm for multi-task learning. Subsequent works extend graph prompts to allow in-context learning without weight update (Huang et al., 2023a; Galkin et al., 2023). However, these works only tackle limited types of tasks and do not generalize to new domains. Hence, researchers propose integrating LLM into the graph learning.

**GNNs and Transformers:** In recent years, GNNs have become the most popular method for dealing with graph learning problems due to their extraordinary ability in structural learning. Particularly, Previous works (Xu et al., 2018; Morris et al., 2019) show that the expressive power of message-passing GNNs can be as powerful as the 1-dimensional Weisfeiler-Lehman test, a powerful algorithm for graph isomorphism problems. Many recent works also try to design more powerful GNNs that beyond the 1-WL test (Zhang & Li, 2021; Kong et al., 2022; Feng et al., 2022; Huang et al., 2023b; Zhang et al., 2023b;a; Feng et al., 2023; Kong et al., 2023) for better structural ability like learning distance between nodes or counting cycles in graph. Some works try to combine the GNN with the transformer. particularly, GraphFormers (Yang et al., 2021) and GROVER (Rong et al., 2020) also insert a GNN layer between consecutive transformer layers for modeling graph inductive bias. Different from us, their transformer layers are randomly initialized and directly tuned on downstream tasks without text.

Table 10: Comparison of LLM-generated answers with correct answers for graph structure questions.

| Question | Answer | Correct Answer |
|---|---|---|
| In the graph node 8 connects to 1, node 13 connects to 9, node 8 connects to 7, node 3 connects to 13, node 3 connects to 4, node 6 connects to 6, node 11 connects to 8, node 12 connects to 11, node 2 connects to 12, node 14 connects to 7, node 6 connects to 3, node 6 connects to 9, node 2 connects to 8, node 2 connects to 9, node 5 connects to 8, node 3 connects to 11, node 8 connects to 5, node 10 connects to 11, node 2 connects to 9, node 9 connects to 10, what is a shortest path between node 8 and node 9? | To find the shortest path between node 8 and node 9, we can use Dijkstra's algorithm or any other shortest path algorithm. However, since this graph is small, we can find it manually. Here's one of the shortest paths between node 8 and node 9: Start at node 8. Move from node 8 to node 5 (neighbor). Move from node 5 to node 8 (neighbor). Move from node 8 to node 2 (neighbor). Move from node 2 to node 9 (neighbor). So, the shortest path between node 8 and node 9 is: **8 -> 5 -> 2 -> 9**. | **8 -> 2 -> 9** |
| In the graph node 8 connects to 1, node 13 connects to 9, node 8 connects to 7, node 3 connects to 13, node 3 connects to 4, node 6 connects to 6, node 11 connects to 8, node 12 connects to 11, node 2 connects to 12, node 14 connects to 7, node 6 connects to 3, node 6 connects to 9, node 2 connects to 8, node 2 connects to 9, node 5 connects to 8, node 3 connects to 11, node 8 connects to 5, node 10 connects to 11, node 2 connects to 9, node 9 connects to 10, what are common neighbors between 13 and 5? | To find the common neighbors between nodes 13 and 5, we need to identify all nodes that are neighbors of both nodes 13 and 5. From the provided connections: Node 13 is connected to node 9 and node 3. Node 5 is connected to node 8. Therefore, **the common neighbor between node 13 and node 5 is node 8**. | **No common neighbors.** |

# E    ADDITIONAL DISCUSSIONS

## E.1    GRAPH STRUCTURE QUESTION EXAMPLE OF LLM

We assessed the ability of LLMs to respond to questions related to graph structures, including shortest path distances and common neighbor counting. For this evaluation, graph edges were described using plain text, and the LLM was tasked with generating the answers. The results of this evaluation are presented in Table 1. These findings indicate that LLMs struggle to comprehend graph structures effectively. We include examples of the questions posed and the corresponding answers generated by the LLM in Table 10, to illustrate these challenges.

## E.2    THEORETICAL ADVANTAGES OF GOFA'S GRAPH LANGUAGE ENCODER

In GOFA, we innovatively integrate GNN layers into LLMs to help LLMs understand graph structures. This approach is theoretically more powerful and suitable for designing GFMs than using pure LLMs. Graph data have unique properties, such as node permutation invariance without fixed ordering (Keriven & Peyré, 2019), making sequential models like LLMs unsuitable for modeling graphs. For a graph with $n$ nodes, the number of possible orders is $n!$, which means sequential models like LLMs will need factorial sample complexity to learn that all of them correspond to the same graph. Existing works therefore use random or heuristic order to represent a graph, resulting in

suboptimality and poor generalization on structure-related tasks. One empirical example from the task planning experiments of LLM agents in (Wu et al., 2024) shows that LLMs can only perform well on task graphs with *a specific node ordering* but cannot maintain this accuracy after nodes are reordered. Methods like LLaGA (Chen et al., 2024) also fall into this category and are suboptimal if the task requires deep structure understanding. Instead, GNNs are a powerful choice widely accepted by the literature for encoding both features and the structure of graphs. They are permutation equivariant to graph order and can learn invariant structure information. Our GOFA , interleaving GNN layers into LLMs, naturally preserves this property. Specifically, for LLM layers, each node is processed individually, which is obvious that we can keep the permutation invariance. As GNN layers are permutation invariant, this conclude that the GOFA is permutation invariant to input graph.

### E.3   ADVANTAGES OF GOFA'S SELF-SUPERVISED LEARNING TASKS

Our proposed self-supervised learning tasks are enlightened by existing graph and NLP SSL tasks. However, our tasks are novel compared to existing methods from several perspectives.

In graph SSL, most prior work aims to recover original features or graph structure contrastively or generatively (Liu et al., 2022), using learned embeddings for downstream classifiers. In contrast, our tasks aim to learn embeddings that enable downstream natural language generation. Concretely, our SSL shortest path prediction task requires the model to output multiple actual paths (e.g., node a→node b→node c) between two nodes in text format; this requires a more fine-grained and in-depth model understanding of the graph structure. Regular graph SSL tasks such as link prediction and shortest path distance prediction only cares about a simple objective (binary classification for link and single number for distance). Conversely, our generation-oriented design allows a unified task format to query into the graph from different aspects with different granularity (e.g. shortest path and common neighbor can be incorporated under the same natural language generation framework), whereas traditional SSL tasks inevitability lose much detailed information and may need artificial complicated design to accommodate multiple SSL targets.

SSL in NLP (e.g. next-word-prediction) only takes texts as input. One may think of directly converting graphs to text and doing a similar SSL. However, as many previous works show (Wang et al., 2023), converting graphs directly to texts for LLM generation is ineffective. Hence, we design the sentence completion task directly on the graph to use connection information to help the model attend to correct nodes for subsequent generation.

In summary, our SSL design cares more about training the model to generate any answers in natural language format so that it can accommodate arbitrary tasks, which differs from traditional graph SSL that normally focuses on classification/regression (actually any graph SSL tasks can be incorporated into our natural language generation framework). Compared to NLP SSL, our novel SSL design focuses on sentence completion using neighboring sentence information rather than pure auto-regression, strengthening the model's power to leverage joint graph-text information.

## F   EXPERIMENTAL SETTINGS

### F.1   GENERAL SETTINGS

**Subgraph sampling**: In the GOFA , for node/link/graph-level tasks, the input format is unified as a subgraph task. Namely, for node/link-level tasks, we will select a $k$-hop subgraph surrounding the target nodes as the input graph for the model. We follow a similar subgraph sampling strategy as OFA (Liu et al., 2023a). Specifically, for node-level tasks, we directly sample the $k$-hop subgraph rooted at the target node. Meanwhile, we set a threshold for maximum nodes per hop. If the nodes in a certain hop exceed the threshold, we will randomly sample maximum nodes from all nodes. For link-level tasks, we doing the sampling on both two nodes.

**Implementations**. Both the GOFA and all baselines are implemented using Python with Pytorch, transformers, and PyG (Fey & Lenssen, 2019) packages.

## F.2 Design of pre-training tasks

In this section, we describe the self-supervised pre-training of GOFA . The goal of the pre-training is to let the GOFA model obtain the ability to query graph structure and context but retain the ability to reason about plain text. Specifically, we perform the pre-training task using multiple existing graph datasets, including MAG240M (Hu et al., 2021a), Arxiv (Hu et al., 2021b), Pubmed, Wikikg90mv2 (Hu et al., 2021a), and Ultrachat200k (Ding et al., 2023) datasets. Further, we create another graph dataset called WikiGraph, whose graphs are generated from sentences in the pure textual WikiData dataset (Merity et al., 2022) using LLM. Details about the datasets can be found in Appendix C. We randomly generate a unique node ID (such as [Node A]) for each node in the training sample and append it to the original node text. This ID will serve as a basis for querying nodes in the graph. We design four pre-training tasks: sentence completion, structural understanding, question-answering, and information retrieval tasks. Figure 1 shows an example of each task. We describe the rationale of each task below and leave some implementation details to Appendix F. We also include an additional discussion of the advantages of our designed tasks in Appendix E.3.

**Sentence completion task**. The objective of the sentence completion task is to train GOFA to reason about the rest of the text in a node given both the existing text and the information in the rest of the graph. Given an input training sample, we randomly select $n$ nodes in the graph as the target nodes. All selected nodes' texts are split into halves. The first half forms node text $x(v)$, and the second half becomes the target $y$ to generate. The length of the first half will also be randomly determined. Finally, the output representation of these $n$ nodes will be directly input to the decoder (no prompt node will be connected) and we minimize the loss between model decoded text and target $y$. This sentence-completion pre-training task adapts LLMs' standard ability to the graph context.

**Structural understanding tasks**. The objective of the structural tasks is to pre-train GOFA to understand basic graph structural properties. In light of this, we design the shortest paths and common neighbors reasoning tasks between nodes. Specifically, For each training subgraph sample, we randomly sample $n$ node pairs as the selected targets. For each selected node pair, we ask the model to compute the shortest path distance between two nodes and output all possible shortest paths between them using the assigned node IDs. Meanwhile, we also ask the model to output the number of common neighbors the two nodes have and the node IDs of their common neighbors. For the structural understanding task, a prompt node $v_p$ will connect to both two nodes since our structural tasks need the model to reason about two nodes simultaneously. The text in the prompt node will be the corresponding question. Through these two tasks, the model is expected to gain the ability to identify basic graph structures, which are critical to downstream tasks.

**Question answering task**. Unlike language corpus, which naturally contains many question-and-answer (QA) pairs, graph data usually only contain objective descriptions of entities. Nevertheless, for the model to be fluid in tasks, we need the model to understand user prompts and be sensitive to different tasks. Hence, we synthesize a QA-chain dataset from Ultrachat200k, as shown in Figure 1. A language QA sequence is converted to a chain graph where nodes with question texts alternate with nodes with answer texts, which are connected by directed edges to represent the conversation order. The last question becomes the text on prompt node $v_p$, which is connected to every node in the chain, and the last answer is the target text $y$ (see Figure 1 QA-Chain Task for an example). This QA task provides QA pseudo-graphs missing from the common graph corpus, and we found it critical for enabling the model to be responsive to arbitrary tasks expressed in free-form text questions.

**Information retrieval task.** For most of the downstream tasks, GOFA requires a prompt node to link to all target nodes in the graph to solve related problems. To enable the prompt node to effectively maintain related information for solving the task in the decoding stage, we design an information retrieval task to realize these goals. Specifically, for each input graph, we randomly select $n$ nodes and we connect a prompt node to these $n$ nodes. Next, the information retrieval task is further divided into two parts: key-to-content and content-to-key. For key-to-content, we provide a node ID (randomly chosen from the selected $n$ nodes) in the prompt node and ask the model to retrieve the text of that node. For the content-to-key task, we provide the content of one node (selected the same as above) in the prompt node and ask the model to return the correct node ID of that node. This task enhances the ability of GOFA to utilize our provided node IDs to retrieve and maintain correct information in the prompt node, which proves useful for many downstream tasks requiring information retrieval.

Table 11: Detailed question and answer example in pertaining task.

| Task | Question example | Answer example |
|---|---|---|
| Sentence completion | *Complete the sentence of the target node.* 
 *Complete the sentence of the node[NODE.A].* | *The rest of the sentence in the target node.* 
 *The rest of the sentence in node [NODE.A].* |
| Shortest paths | *Compute the shortest path distance between the target node [NODE.L] and node [NODE.B] and generate all shortest paths from the target node to the node [NODE.B]. Please separate nodes in the path with ->. If multiple paths exist, generate all of them with an ascending order of node sequences and separate different paths with ;.* | *The shortest path distance is 2. Shortest paths: [NODEID.L] -> [NODEID.G] -> [NODEID.B].* |
| Common Neighbors | *Is there any common neighbor between the target node [NODE.L] and node [NODE.B]? If it exist, please give the total number and list all common neighbors in ascending order of node, separate nodes with ;.* | *There is 1 common neighbor between two nodes, including [NODEID.G].* |
| QA-Chain | *What are the rules and restrictions in place for COVID-19 in the city?* | *I don't have any live data regarding the covid-19 rules and restrictions. Please check with the local authorities or health department for the latest guidelines and restrictions in your city.* |
| Information Retrieval | *Please output the content of [NODE.A].* 
 *Given this node content: {node content}, please output the node id.* | *Content on [NODE.A].* 

 *[NODEID].* |

### F.3 PRE-TRAIN IMPLEMENTATION DETAILS OF GOFA

**Dataset and task construction**. As we discussed, we designed four different pre-training tasks for GOFA . Here we describe some implementation details about each task and then discuss how we construct each task on each dataset.

For the sentence completion task, the node text is split by the following rule: for each node, if the node text is less than 256 words, we set the maximum left-halve length to be the half of node sentence length. Otherwise, we set it to 128. Next, we randomly choose a length from 0 to maximum left length as the final cut point to cut the sentence into two pieces. For the shortest path task, we ask the model to output both the shortest path distance and all possible shortest paths. Since there may be multiple paths, to ensure the uniqueness of the answer, we first order all paths based on the node ID (the ascending order of alphabets) for nodes in each path and ask the model to learn this order. The construction of common neighbor task is similar. Finally, for information retrieval, given an input graph sample, we randomly select 2 to the number of nodes in the graph to be the target nodes.

For pre-training datasets, we use multiple datasets including MAG240M, Arxiv, Pubmed, Wikikg90mv2, Ultrachat200k, and WikiGraph. For MAG240M, Arxiv, and Pubmed datasets, each

training sample is a subgraph sampled around a node. Next, sentence completion, shortest path, and common neighbor tasks are constructed. For each sample, there are 4 complete sentences, 3 shortest path, and 3 common neighbor tasks. We will also construct information retrieval tasks on these datasets. However, to ensure a moderated graph size, the information retrieval task will be constructed separately from the above tasks and also for both key-to-content and content-to-key tasks. For each information retrieval task sample, there will be only one task. For Wikikg90m, each training sample is a subgraph sampled around an edge. In Wikikg90m, we additionally include a link prediction task. That is, for each input graph, we randomly mask $e$ edges and ask the model to recover the content in the edge. For each sample, there are 4 complete sentences, 2 shortest paths, and 2 common neighbor tasks, and 2 link prediction tasks. At the same time, the information retrieval task will also be generated separately. For WikiGraph, each sample is itself a graph. Similar to Wikikg90mv2, each sample consists of 4 complete sentences, 2 shortest paths, 2 common neighbor tasks, and 2 link prediction tasks and information retrieval task will also be generated separately. Finally, for Ultracha200k, we only include question answer task and each sample only contains one task. The detailed task prompts and answer examples are shown in Table 11.

**Training details**. The initial weight of the LLM compressor and decoder is obtained from ICAE (Ge et al., 2023). The initial weight of all GNN layers is randomly initialized. The value of all gates in the residual connection is set to 0 to ensure the initialized model performs the same as the original language model. During the training, we only tune the GNN layers. For each training epoch, the training corpus includes 500,000 MAG240M samples, 50,000 Arxiv samples, 5,000 PubMed samples, 100,000 Ultrachat200k samples, 80,000 WikiGraph samples, 100,000 Wikikg90mv2 samples. At the meantime, for MAG240M, Arxiv, Pubmed, WikiGraph, and Wikikg90mv2, we will include 10,000 key-to-content and 10,000 content-to-key information retrieval tasks. This resulted in 935,000 samples for each training epoch and we trained the model for 3 epochs. The training is conducted on 8 NVIDIAA100_SXM4_80GB GPUs with DeepSpeed stage 2 (Rajbhandari et al., 2020) parallelism. The detailed training parameters are set the same for both two models and are listed in Table 12. We use AdamW optimizer with $\beta = (0.9, 0.95)$. We use a cosine annealing learning rate scheduler, and the minimum learning rate is 10% of the initial learning rate. We restarted the learning rate 2 times on one-third and two-thirds of the training.

Table 12: Hyper-parameters for pretraining.

| lr | weight_decay | batch_size | dropout | grad_clip | gradient_accum | llm_max_length | optimizer |
|---|---|---|---|---|---|---|---|
| 0.0001 | 0.1 | 8 | 0.0 | 0.5 | 8 | 128 | AdamW |

## F.4 ZERO-SHOT LEARNING

**Setting**. For the zero-shot learning, we select Cora-link, Cora-node, WikiCS, Products, ExplaGraphs, and SceneGraphs as evaluation datasets. For all datasets, we directly evaluate baselines and GOFA on the test set.

**Baseline Details**: We compare the performance of GOFA with two categories of baseline methods. The first category includes models that directly utilize large language models (LLMs). For this, we select Llama2-7B and Mistral-7B (Jiang et al., 2023) as baselines. We input the content of all target nodes into these pre-trained models and concatenate the same prompt used in GOFA for evaluation. The second category consists of Graph LLM models that have zero-shot ability. We include OFA (Liu et al., 2023a), GraphGPT (Tang et al., 2023), UniGraph (He & Hooi, 2024), ZeroG (Li et al., 2024), and LLaGA (Chen et al., 2024) as baselines. For OFA, we extend the datasets by adding Products and ExplaGraphs and follow the original source code to train the model on Arxiv and FB15K237 for 30 epochs, using Llama2-7B as the embedding model. All other settings remain consistent with the default OFA configuration, and we report the test performance accordingly. For GraphGPT, we use the results reported in the LLaGA paper. For UniGraph, we use the results from the original paper. For ZeroG, we use the results in UniGraph paper. For LLaGA, we rerun the source code, adapting the settings of ways to align with our experimental setup.

**Detail of GOFA**. For the GOFA, we fine-tune the model from the pre-training checkpoint. In fine-tuning, we will train the parameters of GNN and LoRA layers in the LLM decoder. To com-

Table 13: Prompt examples of GOFA for each training dataset in Zero-shot learning.

| Dataset | Prompt |
|---------|--------|
| MAG240M | *This is a citation network from microsoft academic graph platform. Nodes represent academic papers and edges represent citation relationship. You are an expert in computer science. You need to choose the correct paper category based on the paper content and its citation network. For example, if the paper [NODEID] {<label_description>, choose <label>;}. What is the most likely paper category for the target paper? Choose from the following: {<label>}.* |
| Pubmed-link | *This is a co-citation network from the Pubmed platform focusing on diabetes mellitus. Nodes represent academic papers and edges represent two papers that are co-cited by other papers. You are a diabetes mellitus expert tasked with determining whether two given papers [NODEID1] and [NODEID2] are co-cited by another paper based on their content and network characteristics. Evaluate the following criteria: assess whether the topics of the two papers are similar, check if the shortest path distance between the two papers is small, and verify whether the papers have a large number of common neighbors in the citation network. If the answer to most of these questions is Yes, choose Yes; if the answer to most of these questions is No, choose No.* |
| PubMed-node | *This is a co-citation network from the Pubmed platform focusing on diabetes mellitus. Nodes represent academic papers and edges represent two papers that are co-cited by other papers. You are an expert on diabetes mellitus. You need to choose the correct paper category based on the paper content and its co-citation network. For example, if the paper [NODEID] {<label_description>, choose <label>;}. What is the most likely paper category for the target paper? Choose from the following: {<label>}.* |
| Wikikg90m | *This is a graph extracted from the entire Wikidata knowledge base. You are an expert in knowledge graph reasoning. You need to choose the correct relation type between two target entities based on their existing relations. For example, if two relations involve {<label_description>, choose <label>;}. What is the relationship between two target entities? Choose from the following list: {<label>}."* |
| Arxiv_node | *This is a citation network from Arxiv platform focusing on the computer science area. Nodes represent academic papers and edges represent citation relationships. You are an expert in computer science. You need to choose the correct paper category based on the paper content and its citation network. For example, if the paper [NODEID] {<label_description>, choose <label>;}. What is the most likely paper category for the target paper? Choose from the following: {<label>}.* |
| Arxiv_link | *This is a citation network from Arxiv platform focusing on the computer science area. Nodes represent academic papers and edges represent citation relationships. You are a computer science expert tasked with determining whether two given papers [NODEID1] and [NODEID2] are co-cited by another paper based on their content and network characteristics. Evaluate the following criteria: assess whether the topics of the two papers are similar, check if the shortest path distance between the two papers is small, and verify whether the papers have a large number of common neighbors in the citation network. If the answer to most of these questions is Yes, choose Yes; if the answer to most of these questions is No, choose No.* |

Table 14: Prompt examples of GOFA for each evaluation dataset in Zero-shot learning.

| Dataset | Prompt |
|---|---|
| Cora-node | *This is a co-citation network focusing on artificial intelligence, nodes represent academic papers and edges represent two papers that are co-cited by other papers. You are an expert in computer science. You need to choose the correct paper category based on the paper content and its co-citation network. For example, if the paper [NODEID] {<label_description>, choose <label>;}. What is the most likely paper category for the target paper? Choose from the following: {<label>}.* |
| Cora-link | *This is a co-citation network focusing on artificial intelligence, nodes represent academic papers, and edges represent two papers that are co-cited by other papers. You are a computer science expert tasked with determining whether two given papers are co-cited by another paper based on their content and network characteristics. Evaluate the following criteria: assess whether the topics of the two papers are similar, check if the shortest path distance between the two papers is small, and verify whether the papers have a large number of common neighbors in the citation network. If the answer to most of these questions is Yes, choose Yes; if the answer to most of these questions is No, choose No.* |
| WikiCS | *This is a Wikipedia graph focusing on computer science. Nodes represent Wikipedia terms and edges represent two terms that have hyperlinks. You are an expert in computer science. You need to choose the correct category of Wikipedia term based on the term content. For example, if the term [NODEID] {<label_description>, choose <label>;}. What is the most like category for this Wikipedia entry? Choose from the following: {<label>}.* |
| Products | *This is a co-purchase network from the Amazon platform. Nodes represent the products sold on Amazon and edges represent two products that are co-purchased together. For example, if the product [NODEID] {<label_description>, choose <label>;}. What is the most like category for this product? Choose from the following: {<label>}.* |
| FB15K237 | *This is a knowledge graph from the FreeBase. Nodes represent knowledge entities and edges represent relations between two entities. You are an expert in knowledge graph reasoning. You need to choose the correct relation type between two target entities based on their existing relations. For example, if two relations {<label_description>, choose <label>;}. What is the relationship between two target entities? Choose from the following list: {<label>}."* |
| ExplaGraphs | *This is a graph constructed from commonsense logic. Nodes represent commonsense objects and edges represent the relation between two objects. You are a logic expert tasked with analyzing the logical relationship between two arguments related to connected entities. Determine if the arguments support or counter each other based on their logical coherence. If there is no logical conflict between the two arguments and they are in agreement, choose Support; if the arguments exhibit a logical conflict or contradiction, choose Counter.* |
| SceneGraphs | *This is a scene graph generated from an image. Nodes represent an object in the image and edges represent the relationship between two objects. <Question>* |

Table 15: Hyper-parameters for zero-shot instruction fine-tuning.

| lr | weight_decay | gradient_accum | llm_max_length |
|---|---|---|---|
| 0.0001 | 0.1 | 64 | 256 |

prehensively evaluate the performance of GOFA, We separately fine-tune the GOFA on different datasets. Specifically, we design two different settings. In the first setting, we fine-tune the model using the Arxiv and Pubmed datasets with both the node classification and link prediction tasks. In the second setting, we add mag240m and Wikikg90m additionally. We denote GOFA-T and GOFA-F, respectively. For GOFA-T, we sample 40000, 80000, 10000, 10000 for Arxiv_link, Arxiv_node, Pubmed_link, and Pubmed_node, respectively. For GOFA-F, we sample 10000, 10000, 40000, 50000, 10000, 10000 for MAG240M, Wikikg90m, Arxiv_link, Arxiv_node, Pubmed_node, and Pubmed_link, respectively. For all evaluation and pre-training datasets, we design multiple prompt templates with instructions to let the model select the correct label from the provided label list. For each label in each dataset, we use the GPT-4 to generate a short description for the label. The detailed prompt examples for all datasets are shown in Table 13 and Table 14. For all MAG240M, Wikikg90m, and Arxiv, since it is hard to include all ways in the prompt, we randomly sampled 10 ways during the training for each sample. For each pre-training dataset, we randomly sample a fixed number of training samples in a random way. The detailed parameters for fine-tuning are listed in Table 15. All parameters not listed in the table are the same as the pre-training setting. For all training versions, we directly evaluate the model on the test set of all evaluation datasets. We evaluate the model on the test set. For datasets with less than 15000 test samples, we evaluate on the whole set. Otherwise, we only randomly select 15000 samples for evaluation, due to the time constraint. For evaluation, we will match the text output generated by the GOFA with the ground true label to compute the accuracy of the classification task. For the regression task, we will extract the number from the output text and compute the metric with the correct value.

## F.5 Supervised-learning

Table 16: Hyper-parameters for supervised fine-tuning.

| lr | weight_decay | grad_clip | gradient_accum | llm_max_length |
|---|---|---|---|---|
| 0.0001 | 0.1 | 0.5 | 32 | 256 |

**Setting**. For the supervised-learning setting, we select Cora (node/link), PubMed (node/link), Arxiv, WikiCS, WN18RR, FB15K237, and Products datasets for the evaluation. For all datasets, we utilize the default split described in Appendix C. To ensure a fair comparison, we employ subgraph sampling for GOFA and all baseline methods. For all datasets, the sampling hop is 3 and the maximum nodes per hop are 5.

**Detail of baselines**. For the traditional GNN methods, we include GCN (Kipf & Welling, 2017) and GAT (Veličković et al., 2018). To ensure a fair comparison, we use Llama2-7B to convert raw texts in all datasets to sentence embedding and use this as the model's input node/edge features. We re-implement both methods in order to adapt the original method with subgraph input. Specifically, for but node/link-level tasks, we will add labeling trick (Zhang et al., 2021) to the target nodes at the beginning. After message passing, we will use the summation pooling on all target nodes and use the result embedding for the prediction. For traditional GNN methods, we train and evaluate each dataset independently. For all datasets, we search the number of layers and dropout parameters. For each parameter set, we repeat the experiment 4 times select the parameter set with the best validation performance, and report the performance on the test set. For constrastive learning methods, we include DGI (Veličković et al., 2018) and BGRL (Thakoor et al., 2021). We directly report results from UniGraph (He & Hooi, 2024) . For the graph foundation model, we include OFA (Liu et al., 2023a) and UniGraph (He & Hooi, 2024) as the baseline. The OFA is simultaneously trained and evaluated on all datasets. To ensure a fair comparison, we get their code from the original source and train the model on Cora (node/link), PubMed (node/link), Arxiv, WikiCS, WN18RR, and FB15K237 dataset using the Llama2-7b as base LLM model. Similarly, for OFA, we use the

Table 17: Detailed prompt of GOFA for each dataset in supervised learning.

| Dataset | Prompt |
|---------|--------|
| Cora-node | *This is a co-citation network focusing on artificial intelligence, nodes represent academic papers and edges represent two papers are co-cited by other papers. What is the most likely paper category for the target paper? Please directly answer the category.* |
| Cora-link | *This is a co-citation network focusing on artificial intelligence, nodes represent academic papers and edges represent two papers are co-cited by other papers. Is the two target papers co-cited or not? Please only answer yes or no.* |
| PubMed-node | *This is a co-citation network from Pubmed platform focusing on diabetes mellitus. Nodes represent academic papers and edges represent two papers are co-cited by other papers. What is the most likely paper category for the target paper? Please directly answer the category.* |
| PubMed-link | *This is a co-citation network from Pubmed platform focusing on diabetes mellitus. Nodes represent academic papers and edges represent two papers are co-cited by other papers. Is the two target papers co-cited or not? Please only answer yes or no.* |
| Arxiv | *This is a citation network from arxiv platform focusing on the computer science area. Nodes represent academic papers and edges represent citation relationships. What is the most likely paper category for the target Arxiv paper? please directly answer the category.* |
| WikiCS | *This is a Wikipedia graph focusing on computer science. Nodes represent Wikipedia terms and edges represent two terms have hyperlink. What is the most likely category for this Wikipedia term? Please directly answer the category.* |
| WN18RR | *This is a knowledge graph from WordNet. Nodes represent an English word and edges represent the relationship between two words. What is the relationship between two target words? Please directly answer the relationship.* |
| FB15K237 | *This is a knowledge graph from freebase. Nodes represent knowledge entities and edges represent relations between two entities. What is the relationship between two target entities? Please directly answer the relationship.* |
| Products | *This is a co-purchase network from the Amazon platform. Nodes represent the products sold on Amazon and edges represent two products are co-purchased together. What is the most like category for this product? Please directly answer the category.* |

same subgraph sampling parameters as all other methods. For other parameters, we use the default parameter provided in their code. We only run the model one time and report the final performance. For UniGraph, we directly report results from their original paper.

**Detail of GOFA**. For the GOFA, we fine-tune the model from the pre-training checkpoint. In fine-tuning, we will train the parameters of GNN and LoRA layers in the LLM decoder. We simultaneously fine-tune the model on the train set of Cora-node, Cora-link, PubMed-node, PubMed-link, Arxiv, WikiCS, WN18RR, FB15K237, and Products. For each dataset, we will randomly sample a fixed number of training samples for each epoch with random sampling. The sample numbers for each dataset is 3000, 40000, 3000, 80000, 105000, 12000, 60000, 120000, and 38000, respectively. We fine-tune the model for 1 epochs. The detailed parameters for fine-tuning are listed in Table 16. For each dataset, we create a prompt for the LLM decoder to generate the desired answer. In a supervised setting, we ask the LLM model directly to generate the correct answer, instead of doing the selection from the given list. The detailed prompt for each dataset is listed in Table 17. For evaluation, we will

match the text output generated by the GOFA with the ground true label to compute the accuracy of the classification task. We evaluate the model on the test set. For datasets with less than 15000 test samples, we evaluate on the whole set. Otherwise, we only randomly select 15000 samples for evaluation, due to the time constraint.

