# OpenReview forum: "GOFA: A Generative One-For-All Model for Joint Graph Language Modeling"
_ICLR.cc/2025/Conference — ICLR 2025 Poster_

### Official Review · Reviewer_5YVL · 2024-10-29

**Soundness:** 3
**Presentation:** 4
**Contribution:** 3
**Rating:** 6
**Confidence:** 4

**Summary:**

This paper introduces three key properties that an effective Graph Foundation Model (**GFM**) should possess: self-supervised pretraining, fluidity, and graph understanding. To meet these goals, the authors propose Generative One-for-All (**GOFA**) , which integrates a Language Model Decoder with a Graph Language Encoder, which strategically interleaves trainable Graph Neural Network (GNN) layers into frozen Large Language Model (LLM) layers.

The GOFA model is pre-trained on a diverse set of datasets, including MAG240M, Pubmed and Arxiv, Wikikg90mv2, WikiGraph, and Ultrachat200k. Notably, the WikiGraph dataset is created specifically for this work and it is derived from the existing WikiText dataset.

The pretraining results validate the effectiveness of incorporating GNN layers into LLMs, while instruction-tuning experiments further demonstrate GOFA's strong zero-shot generalization capabilities.

**Strengths:**

The paper is well-structured and easy to follow. Its key strengths can be summarized as follows:

1. The authors identify three crucial properties of an effective graph foundation model: self-supervised pretraining, fluidity, and graph understanding. I strongly agree with these as important considerations.

2. The paper introduces GOFA, which operates under a graph completion framework to enable large-scale self-supervised learning. Furthermore, GOFA demonstrates impressive performance in both zero-shot and fine-tuning scenarios when compared to LLM baselines.

3. One potential concern in developing a foundation model is whether it might negatively impact the capabilities of existing LLMs. However, the results in Table 1 suggest that GOFA's training does not degrade the performance of the original pre-trained LLMs. This is a critical design criterion when building a graph foundation model.

**Weaknesses:**

1. The authors suggest that integrating GNN layers into LLMs offers a theoretical advantage over standard LLMs when it comes to modeling graph structures. However, I did not find any accompanying mathematical formulations or equations to substantiate this claim in the appendix.

2. GOFA performs significantly worse than the LLM baselines on the SceneGraphs dataset, despite the authors attributing this to the instruction-tuning data containing information-dense texts. It's concerning that a domain-specific foundation model, despite careful design, does not demonstrate competitive performance.

3. As the authors note, using a frozen LLM compressor while tuning only the GNN layers appears questionable. This approach might impede optimal alignment between information from various modalities, potentially limiting the model's overall effectiveness.

**Questions:**

I have two questions for the authors:

1. Dataset quality: I would like to inquire whether the authors thoroughly validated the content generated in their constructed WikiGraph dataset. Given that the dataset is created by prompting large language models (LLMs), there is a known issue with the potential for LLMs to produce hallucinated or inaccurate outputs. Could the authors elaborate on the measures they took to ensure the accuracy and quality of the dataset? How did they mitigate the risks of including erroneous or noisy data in the final dataset?

2. Could the authors provide details about the pre-training time when using 8 NVIDIA A100-SXM4-80GB GPUs?

---

> ### Author Response · Authors · 2024-11-19
> **Response to Reviewer 5YVL (1/2)**
>
> >W1: The authors suggest that integrating GNN layers into LLMs offers a theoretical advantage over standard LLMs when it comes to modeling graph structures. However, I did not find any accompanying mathematical formulations or equations to substantiate this claim in the appendix.
> >
>
> Thanks for the advice. Here we provide additional discussion on the theoretical advantage. Define $g \in \mathcal{G}$ as the permutation operation in the permutation group $\mathcal{G}$. We say a function is permutation invariant if $\forall g \in \mathcal{G}$, we have:
>
> $$
> f(x) = f(gx).
> $$
>
> In graph, a typical permutation operation $g$ can permute the order of nodes in the graph. Let a graph input be $G=(V, E)$, where $V$ is the set of nodes and $E$ is the set of edges. We denote a permutation operation on the graph as $gG=(gV, gE)$. As for LLM, the graph is first flattened and then input into LLM with positional encoding on each token or token pair, it is easy to prove that:
>
> $$
> \text{LLM}(\text{flatten}(V, E)) \neq \text{LLM}(\text{flatten}(gV, gE)), \forall g\in \mathcal{G}
> $$
>
> Instead, GNNs are designed with permutation invariant in mind and naturally hold that $\text{GNN}(G) = \text{GNN}(gG), \forall g\in \mathcal{G}$. Finally, we show that GOFA that integrate GNN into LLM are permutation invariant to node order.  In LLM layer, each node and edge are fed into the model independently. That is:
>
> $h^{l+1}_{v}$ $=$ $\text{LLM}_l$ $(h_v^l)$ $,\forall v \in V,$
>
> and
>
> $h^{l+1}_{e}$ $=$ $\text{LLM}_l$ $(h_e^l)$ $,\forall e \in E.$
>
> The permutation of the graph will not affect the process of each individual node or edge. Thus, the permutation invariant holds for LLM layers. For GNN layers, the permutation is held as we discussed above, which concludes that the GOFA is permutation invariant to the node order of the graph.
>
> Finally, we can directly leverage Theorem 7 in [1] to show the generalization gap between LLM and GOFA in modeling graph related problems.  The Theorem is as follows:
>
> Let $\mathcal{X}=\mathbb{R}^d$ denote the input space and $\mathcal{X} \sim \mathcal{N}(0, \sigma^2_{\mathcal{X}}I)$, $\mathcal{Y} = \mathbb{R}$ be the output space. Let $\mathcal{G}$ be a compact group with an orthogonal representation $\phi$ on $\mathcal{X}$. Let the function we want to predict be $Y=h_{\theta}(X) + \xi$ where $h_{\theta}(x)=\theta^Tx$ is $\mathcal{G}$-invariant with $\theta\in \mathbb{R}^d$ and $\xi$ is the noise factor with mean 0, variance $\sigma^2_{\xi}< \infty$ and is independent of $X$. Let $w$ and $\hat{w}$ be the least-squares estimate of $\theta$ from i.i.d examples $\{(X_i, Y_i)|i=1,...,n\}$
> with respect to model with and without $\mathcal{G}$-invariant property . let $A$ be the orthogonal complement of the subspace of $\mathcal{G}$-invariant linear predictors. If $n > d+1$, the generalization gap $\Delta(h_w, h_{\hat{w}})$ can be measured by:
>
> $$
> \mathbb{E}[\Delta(h_w, h_{\hat{w}})]=\sigma^2_{\xi}\frac{\text{dim} A}{n-d-1},
> $$
>
> where $\text{dim}A$  is a measurement of how significant the symmetry is to the problem. A larger $\text{dim}A$  will result in a larger generalization gap and indicate the importance of using a $\mathcal{G}$-invariant function and weight $w$. We refer the reviewer to [1] for more details of the Theorem.
>
> Since the permutation group is a special case of a compact group, we can directly replace the $h_w$ with GOFA and $h_{\hat{w}}$ with LLM, and the theorem still holds. That means if the function $h_{\theta}(x)$ we want to model requires permutation invariant ability, GOFA will have better generalization ability than LLM with a theoretical guarantee. The only thing left is that the real-world data and function we want to model may not be i.i.d and also non-linear, which adds additional complexity to the problem. However, it is natural to expect the gap would be further expanded in such cases. We omit the proof here as it is extremely complex and beyond the scope of our work.

---

> > ### Author Response · Authors · 2024-11-19
> > **Response to Reviewer 5YVL (2/2)**
> >
> > >W2: GOFA performs significantly worse than the LLM baselines on the SceneGraphs dataset, despite the authors attributing this to the instruction-tuning data containing information-dense texts. It's concerning that a domain-specific foundation model, despite careful design, does not demonstrate competitive performance.
> > >
> >
> > Thank you for pointing this out. The SceneGraphs dataset differs significantly from the instruction-tuning datasets used in our model training. Specifically, SceneGraphs is derived from image data, and its tasks often focus on node attributes and relationships between nodes. To answer these questions effectively, the model must identify the relevant nodes in the graph and extract information from their features or the edge features connecting them. This type of knowledge is distinct from the tasks in our instruction-tuning setup, which focuses on node classification and link prediction. As a result, our model did not achieve competitive performance on SceneGraphs.
> >
> > To address this gap, we introduce a new instruction-tuning dataset based on the Visual Genome[2] image dataset. This dataset is tailored to tasks that require reasoning over node attributes and relationships, similar to the SceneGraphs dataset. We fine-tuned our pre-trained model using this new dataset and observed significant improvements with varying amounts of training data. The results are presented in the table below:
> >
> > |  | Instruction-tuning sample size | Scene_graph |
> > | --- | --- | --- |
> > | Mistral-7B | - | 45.95 |
> > | GOFA | 0 | 34.06 |
> > | GOFA | 1000 | 42.4 |
> > | GOFA | 3000 | 50.5 |
> > | GOFA | 10000 | 59.5 |
> >
> > The results show that as the sample size of instruction-tuning data increases, GOFA's performance improves substantially. With 1,000 samples, GOFA reaches 42.4%, almost matching the Mistral-7B baseline. The performance surpasses the baselines when more samples are used for tuning the model, demonstrating its capacity to leverage domain-specific instruction tuning for enhanced performance.
> >
> > >W3: As the authors note, using a frozen LLM compressor while tuning only the GNN layers appears questionable. This approach might impede optimal alignment between information from various modalities, potentially limiting the model's overall effectiveness.
> > >
> >
> > Thanks for the great suggestion! Indeed, we can additionally tune the LLM compressor along with GNN layers for better alignment and we have discussed it in the limitation part. Currently, the main obstacle is that we use HuggingFace to implement both the LLM encoder and decoder. However, HuggingFace does not allow two sets of trainable LoRA weight activated at different stages of a forward pass (one in the encoding stage and one in the decoding stage). We will explore your suggestion when a more flexible implementation is available.
> >
> > >Q1: Dataset quality: I would like to inquire whether the authors thoroughly validated the content generated in their constructed WikiGraph dataset. Given that the dataset is created by prompting large language models (LLMs), there is a known issue with the potential for LLMs to produce hallucinated or inaccurate outputs. Could the authors elaborate on the measures they took to ensure the accuracy and quality of the dataset? How did they mitigate the risks of including erroneous or noisy data in the final dataset?
> > >
> >
> > A: We manually check the quality of the generated graph. In our prompt, we enforce the LLM to stick to the provided context as closely as possible to reduce hallucination. On the other hand, the wikigraph dataset is included to enrich the corpus for pre-training, the goal is for the GOFA model to have an understanding of the general context beyond what’s included in the graph dataset, and the wikigraph dataset serves this purpose well.
> >
> > >Q2: Could the authors provide details about the pre-training time when using 8 NVIDIA A100-SXM4-80GB GPUs?
> > >
> >
> > A: For the pre-training, it takes 153 hours. (2,805,000 graph samples with 22,650,000 input sentences, see Appendix F.3 for the detail).
> >
> > Reference:
> >
> > [1] Elesedy, Bryn, et al. “Provably Strict Generalisation Benefit for Equivariant Models”, ICML21.
> >
> > [2] Krishna, Ranjay, et al. "Visual genome: Connecting language and vision using crowdsourced dense image annotations." *International journal of computer vision* 123 (2017): 32-73.

---

> > > ### Author Response · Authors · 2024-11-24
> > >
> > > Dear reviewer 5YVL,
> > >
> > > Thank you once again for your insightful reviews. As the discussion period is coming to a close, we would like to confirm whether our responses have adequately addressed your concerns. To summarize, we have included a theoretical proof demonstrating the benefits of GOFA compared to plain LLMs for structural modeling. Additionally, we conducted new experiments on the SceneGraph dataset, fine-tuned with a more effective task format. We have also provided detailed discussions on the utilization of the frozen LLM compressor, the quality of the WikiGraph dataset, and the pre-training time. We greatly appreciate your constructive feedback and look forward to your thoughts.
> > >
> > > Best,
> > >
> > > Authors

---

> > > ### Comment · Reviewer_5YVL · 2024-11-25
> > > **Response to comment 1&2**
> > >
> > > Thank you for providing additional discussion on the theoretical advantage of integrating GNN layers into LLMs. Regarding Response 2, I’m curious about the performance of Mistral-7B concerning instruction-tuning sample sizes. The authors do not specify the sample size used for Mistral-7B in the presented table. Are the results for Mistral-7B based on zero-shot evaluation?

---

> > > > ### Author Response · Authors · 2024-11-26
> > > >
> > > > Thank you for your question. Yes, the results for Mistral-7B in the table are based on zero-shot evaluation.

---

### Official Review · Reviewer_qxFo · 2024-11-02

**Soundness:** 3
**Presentation:** 2
**Contribution:** 2
**Rating:** 6
**Confidence:** 4

**Summary:**

The authors introduce GOFA, a general-purpose Graph-Language Model, which proposes an efficient but somewhat intricate approach to encoding a graph into a collection of vectors suitable for input into a large language model (LLM). The model is trained on a large dataset collection using substantial computational resources (8 A100 80GB GPUs).

**Strengths:**

GOFA is trained on multiple datasets and it appears to be indeed a general-purpose Large Graph Language Model.

The paper is well-structured, with a comprehensive presentation of both quantitative and qualitative results across the main text and appendix.

**Weaknesses:**

1. **Evaluation of Graph Structure Utilization.** In Figure 1 and Section 3.4, the authors outline four pre-training tasks: (a) sentence completion, (b) question answering, (c) structural understanding, and (d) information retrieval. However, apart from the structural understanding task, it is somewhat unclear how these tasks benefit from graph information. For instance, tasks like sentence completion and question answering might be achievable by directly providing the raw text input to a generic LLM, possibly yielding similar outcomes. Similarly, in Figure 2, the sentence completion and question-answering tasks are primarily language-oriented and may not require graph-based augmentation. Even in Figure 3, the abstract completion task appears somewhat independent of the graph structure.
\
It would be helpful if the authors could include a comparison where an LLM alone performs these tasks, highlighting any areas where it falls short. Alternatively, if GOFA matches or surpasses the LLM’s performance while adding distinct capabilities, this would serve as a compelling justification for GOFA’s design. Currently, the experimental section primarily focuses on comparing GOFA to LLMs in the structural understanding task (Q3), where graph augmentation evidently improves performance. Additionally, introducing more graph-relevant questions, as seen in recent works (e.g., Fatemi et al.), could further substantiate the advantages of incorporating graph structure in the evaluations.

2. **Comparison with Related Work.** The proposed approach bears similarities to recent models such as GraphToken (https://arxiv.org/abs/2402.05862) and GraphLLM (https://arxiv.org/abs/2310.05845). A detailed comparison, particularly of the advantages and limitations of GOFA relative to these methods, would strengthen the discussion. Including an experimental comparison could also help clarify GOFA’s unique contributions and situate it within the broader context of graph-enhanced LLM research.

**Questions:**

1. Could the authors clarify how the NOG is represented in practice? Specifically, is a special token used for this purpose?

2. Could the authors provide more details about the graph samples used? How many nodes on average are processed by GOFA on each forward pass and how capable is it to provide accurate answers as the graph size increases?

3. How many tokens are reserved for the graph input in the final question-answering LLM (purple tokens in Fig. 3)? Is it a fixed number or does it depend on the graph's size?

---

> ### Author Response · Authors · 2024-11-18
> **Response to Reviewer qxFo (1/3)**
>
> > W1-1: Evaluation of Graph Structure Utilization. In Figure 1 and Section 3.4, the authors outline four pre-training tasks: (a) sentence completion, (b) question answering, (c) structural understanding, and (d) information retrieval. However, apart from the structural understanding task, it is somewhat unclear how these tasks benefit from graph information. For instance, tasks like sentence completion and question answering might be achievable by directly providing the raw text input to a generic LLM, possibly yielding similar outcomes. Similarly, in Figure 2, the sentence completion and question-answering tasks are primarily language-oriented and may not require graph-based augmentation. Even in Figure 3, the abstract completion task appears somewhat independent of the graph structure.
> >
>
> We appreciate the reviewer’s insightful feedback and would like to clarify the broader motivation and necessity behind our pre-training tasks. The primary goal of these tasks—graph sentence completion, question answering, structural understanding, and information retrieval—is not solely to enable GOFA to perform well on these tasks or to use graph information to enhance them. Instead, these tasks serve a more significant purpose: enriching the model's understanding of general graph knowledge and leveraging this enriched understanding to improve downstream **graph tasks**.
>
> For example, the graph sentence completion task allows the model to query information from graph neighbors, enhancing its graph context understanding and resulting in better performance compared to pure LLMs, as demonstrated in Table 1, where GOFA with graph inputs achieves lower perplexity than both LLMs and GOFA with single-node inputs. We would also like to provide some intuition why graph information helps using the sentence completion task as an example. For target nodes, we randomly split its text into two parts, where the first part (can be of length as small as 0) and the neighbors’ full text are used to predict the second part. Consider citation network, where a node represents an article. Without information from its citing articles, it is hard to predict its title accurately, while with graph information, the model can infer the subject, topic, method, etc. much more accurately based on the vast amount of related articles in its neighborhood.  This example intuitively demonstrates the usefulness of graph information for sentence completion.
>
> Similarly, the question answering task trains GOFA to handle free-form queries by utilizing graph context, enabling it to generalize to diverse tasks. For instance, as shown in Figure 7, GOFA successfully outputs the most common product category in a graph even when such tasks were not explicitly seen during training stages. Furthermore, the information retrieval task significantly boosts the model’s generalization ability on graph-based tasks, as evidenced by the results in Figure 5, where pre-training on this task improves GOFA’s zero-shot performance on FB15K237, Products, and Cora-link datasets.
>
> Overall, these tasks collectively enable GOFA to outperform LLMs on language-oriented tasks when graph information is included, while also ensuring the model’s robust transferability and effectiveness on unseen graphs and downstream graph-related tasks.
>
> >W1-2: It would be helpful if the authors could include a comparison where an LLM alone performs these tasks, highlighting any areas where it falls short. Alternatively, if GOFA matches or surpasses the LLM’s performance while adding distinct capabilities, this would serve as a compelling justification for GOFA’s design. Currently, the experimental section primarily focuses on comparing GOFA to LLMs in the structural understanding task (Q3), where graph augmentation evidently improves performance. Additionally, introducing more graph-relevant questions, as seen in recent works (e.g., Fatemi et al.), could further substantiate the advantages of incorporating graph structure in the evaluations.
> >
>
> For all zero-shot evaluations (Q2), we include pure LLMs, LLama2-7B, and Mistral-7B, as baselines (see Table 2 and Table 3). We can see that for most of datasets, GOFA achieves better results than LLMs. Particularly, Cora-node dataset asks the model to classify the category of the paper; WikiCS asks the model to classify the category of a Wikipedia page; Product asks the model to select the best product category. For all these tasks, only providing LLM with the context of the node can already achieve great results. However, by adding graph context, GOFA can further improve the performance. For tasks like Cora-link, where the graph structure are critical for reasoning the existence of edge, LLM shows only about random guess performance. Instead, GOFA achieves much higher performance than LLMs, showing the great graph structure understanding ability of GOFA. All these results demonstrate the distinct capability of GOFA compared to LLMs.

---

> > ### Author Response · Authors · 2024-11-18
> > **Response to Reviewer qxFo (2/3)**
> >
> > > W2: Comparison with Related Work. The proposed approach bears similarities to recent models such as GraphToken and GraphLLM. A detailed comparison, particularly of the advantages and limitations of GOFA relative to these methods, would strengthen the discussion. Including an experimental comparison could also help clarify GOFA’s unique contributions and situate it within the broader context of graph-enhanced LLM research.
> > >
> >
> > Thanks for mentioning these interesting works. GraphToken first uses GNN to encode graphs and concatenate the output tokens with the token representation of questions to input to a frozen LLM for answering questions related to graphs. The key advantage of GOFA over GraphToken is that (1) GraphToken’s graph embedding module is independent of the user query. The graph embedding process might not capture the most query-related information. (2) GraphToken only uses a single vector as node representation which may not contain all information about a node for downstream tasks. On the other hand, GraphToken’s node embedding is much more lightweight, giving it more optimal training and inference time. We will continue exploring the trade-off between information-richness and efficiency in the future.
> >
> > GraphLLM first uses an encode-decoder module to extract textual embedding for each node given query. Next, a graph transformer is employed on graph and textual embedding to learn the graph and output graph representation. Finally, the graph representation is projected to serve as prefixed tokens for each layer of LLMs for answering questions. This method is more similar to GOFA. However, for all these methods, authors mainly conduct **supervised** experiments on synthetic graphs with traditional graph tasks like substructure counting or shortest Path. It is unknown whether these methods can work well or not on real-world graph challenges in a zero-shot setting.
> >
> > We will include all discussions in the updated version.
> >
> >
> > >Q1: Could the authors clarify how the NOG is represented in practice? Specifically, is a special token used for this purpose?
> > >
> >
> > A: The NOG is not a special token but nodes in the graph. Typically, NOG can be any node in a graph given different purposes or tasks. once the representation for each node in the graph is output from GOFA encoder, the representation of NOG will be input into the decoder, along with the optional query, to decode the rest part of a sentence. In our practical training, for sentence completion, we randomly select $k$ nodes from each graph to serve as NOG, and input the representation of each NOG into the decoder independently for sentence completion. For all other tasks including downstream tasks, we will introduce a prompt node, which connects to all nodes that are related to the query. The text in the prompt node is just the query. Then, after encoding, the representation of the prompt node is used for decoding and answering the question.

---

> > > ### Author Response · Authors · 2024-11-18
> > > **Response to Reviewer qxFo (3/3)**
> > >
> > > >Q2: Could the authors provide more details about the graph samples used? How many nodes on average are processed by GOFA on each forward pass and how capable is it to provide accurate answers as the graph size increases?
> > > >
> > >
> > > A: Thanks for the advice.  First, we provide the statistics for all datasets we used (Avg. #NT means the average number of tokens in an edge, and Avg. #ET means the average number of tokens in a node):
> > >
> > > | Dataset | Avg. #N | Avg. #E | Avg. #NT | Avg. #ET | #G |
> > > | --- | --- | --- | --- | --- | --- |
> > > | Cora | 2,708 | 21,112 | 143.4 | 8.0 | 1 |
> > > | Pubmed | 19,717 | 44,338 | 246.2 | 8.0 | 1 |
> > > | Arxiv | 169.343 | 1,166,243 | 174.7 | 7.0 | 1 |
> > > | WikiCS | 11,701 | 216,123 | 421.2 | 6.0 | 1 |
> > > | Product | 54,025 | 144,638 | 113.9 | 6.0 | 1 |
> > > | FB15K237 | 14,451 | 310,116 | 20.1 | 8.4 | 1 |
> > > | ExplaGraphs | 5.17 | 4.25 | 5.1 | 5.3 | 2,766 |
> > > | SceneGraphs | 19.13 | 68.44 | 20.1 | 9.8 | 100,000 |
> > > | MAG240M | 5,875,010 | 26,434,726 | 152.3 | 11.0 | 1 |
> > > | Ultrachat200k | 3.72 | 2.72 | 143.9 | 9.5 | 449,929 |
> > > | Wikikg90m | 91,230,610 | 1,202,155,622 | 18.99 | 25.70 | 1 |
> > >
> > > For all node-level and link-level tasks, we will sample a subgraph around the target node/link to construct input graph. For most of the datasets, The sample hop is 3 and the maximum node in each hop is 5, resulting in a maximum 15 node (do not include prompt node) for each graph. Here, we provide an ablation study on varying different graph sizes:
> > >
> > > | GOFA-T | cora_link | products (10-way) |
> > > | --- | --- | --- |
> > > | 1, 2 | 77.2 | 80.2 |
> > > | 1, 5 | 80.8 | 82.4 |
> > > | 2, 5 | 81.8 | 82.4 |
> > > | 3, 5 | 82.0 | 82.6 |
> > > | 3, 10 | 84.2 | 82.6 |
> > >
> > > We can see that the number of hops and the number of neighbors have different impacts on different datasets. Larger sampled hops are particularly useful for link tasks, as link tasks usually rely on connectivity information. While a large enough first-hop neighbor size is sufficient for the products dataset in which we care more about first hope neighbors (direct co-purchase).
> > >
> > > >Q3: How many tokens are reserved for the graph input in the final question-answering LLM (purple tokens in Fig. 3)? Is it a fixed number or does it depend on the graph's size?
> > > >
> > >
> > > A: In the LLM decoder, the purple token is just the memory token of NOG (128 tokens), which is fixed regardless of the input graph size. However, one can specify multiple NOGs on a graph to query different parts/aspects of the graph.

---

> > > > ### Author Response · Authors · 2024-11-24
> > > >
> > > > Dear reviewer qxFo,
> > > >
> > > > As the discussion period is approaching its end, we would like to thank you again for your thoughtful reviews, your knowledgeable comments are invaluable to us in improving our work. Meanwhile, we’ve responded to your comments in our previous response. In particular, we’ve discussed the purpose behind the design of the pretraining tasks, illustrated GOFA’s distinction to LLM and other LLM+Graph approaches, and clarified the design and implementation details of GOFA. We hope these resolve your concerns. If you have any more questions or feedback, we are more than happy to answer them.
> > > >
> > > > Best,
> > > >
> > > > Authors

---

> > > ### Comment · Reviewer_qxFo · 2024-11-25
> > > **Response to comment 2/3**
> > >
> > > The main reason I mentioned these works (which are just a few of the ones in the broader area of graph + LLM) is because all these works evaluate their models in graph-related tasks. Although such models are trained on smaller, synthetic models, FOGA should be able to perform comparably on questions such as node degree, substructure count, etc. Although the authors claim that "It is unknown whether these methods can work well or not on real-world graph challenges in a zero-shot setting." which I totally agree with, an evaluation of FOGA on "simple" graph tasks is an important piece that is missing completely from the paper.

---

> > > > ### Author Response · Authors · 2024-11-27
> > > > **Response to follow-up questions (1/2)**
> > > >
> > > > Thank you for the follow-up questions! We are more than happy to answer them one by one:
> > > >
> > > > 1. **From your original comment “However, apart from the structural understanding task, it is somewhat unclear how these tasks benefit from graph information.”**
> > > >
> > > > Thank you so much for your further clarification of your concern! To address it more thoroughly, we have revisited and expanded our explanation of why these language-oriented tasks are essential for our pre-training, thereby strengthening our original response. Additionally, as per your suggestion, we have conducted a perplexity comparison with LLMs when provided with the exact same amount of input information as GOFA. We believe this directly addresses your question regarding the usefulness of incorporating structural modeling ability to aid these tasks.
> > > >
> > > > **Why the language-oriented tasks are necessary for GOFA**?
> > > >
> > > > First, we totally agree with you that *a GNN+LLM model, like GOFA, is not necessary to solve pure language-oriented tasks.* For example, the question-answering task is a pure language task, the QA chain is also synthesized from pure language, and we do not expect GOFA to perform better than a conventional LLM. However, in the **pre-training stage**, “exceeding LLM on these language-oriented tasks” is not the purpose. Instead, these tasks are designed to enable the **randomly initialized GNN module** in GOFA to acquire general graph and language knowledge, empowering it to perform better than LLMs (with flattened graph representations) on **real-world downstream graph tasks**.
> > > >
> > > > In this context, the purpose of these pre-training tasks is not to directly use graph information to enhance language tasks, but is use LLMs to help graph models understand general graph tasks in language format.
> > > >
> > > > Missing any pretraining dataset will lead to a less general and powerful GOFA:
> > > >
> > > > - A pretraining process focused solely on structural tasks would leave the GNN module with no knowledge of science (from datasets like MAG240) or common sense (from QA and WikiGraph datasets). This would hinder its ability to process academic or knowledge graphs effectively.
> > > > - Similarly, omitting QA datasets during pretraining would naturally result in a model unable to answer arbitrary questions about graphs. Note that the question on graphs is not limited to graph structures, and Figures 6 and 7 listed some open-ended questions.
> > > >
> > > > More specifically, only after pre-trained with these language-oriented tasks, the GNN module can understand the content in arbitrary graphs and combine its structural understanding and content understanding to solve the graph problem, which is crucial for generalizability. For example, without pre-training on these language-oriented tasks with a broad range of topics, GOFA cannot generalize to product and FB15K237 datasets which are very different from the instruction-tuning dataset arxiv. The zero-shot experiments in the paper also demonstrate the necessity of including language-oriented pre-training for generalizing to unseen graph tasks.
> > > >
> > > > The perplexity experiment in section 5.1 is not suggesting GOFA models language better than LLMs, but it serves as a sanity check to validate that the GOFA model, a newly proposed architecture, can indeed utilize neighbor information to augment generation.
> > > >
> > > > **Fair comparison between GOFA and LLMs on graph completion**
> > > >
> > > > Meanwhile, your suggestion to compare GOFA with an LLM is very insightful. Hence, we add another baseline by feeding flattened graph (as you described) into frozen Mistral-7b and ask the model to complete the missing sentence with the same experimental setup described in Table 1 of the paper. We denote these new results as "Mistral-7B with Graph". Below, we present the updated perplexity results:
> > > >
> > > > |  | Perplexity |
> > > > | --- | --- |
> > > > | Mistral-7B  | 30.12 |
> > > > | Mistral-7B with graph | 25.46 |
> > > > | GOFA | 21.34 |
> > > >
> > > > Interestingly, we still observe that GOFA achieved a lower perplexity, meaning that GOFA’s generation is more correlated with the actual generation target. To understand this, we observe that the graph encodes the strength of correlation, for example, in the product graph, one-hop neighbors of the target have the strongest correlation and should have a higher impact on the generation, in wikics, the connection density among neighbors nodes also explains whether the target node is strongly correlated with its neighbors. This information can easily be learned by GOFA with a GNN module and reflect this in the generation, while it can be difficult for LLM to learn from their flattened representations.

---

> > > > > ### Author Response · Authors · 2024-11-27
> > > > > **Response to follow-up questions (2/2)**
> > > > >
> > > > > 2. **If the whole argument of your work is that extra information is needed (which can be found from another source like a graph), …**
> > > > >
> > > > > We believe the argument of the work is **beyond “**extra information is needed”, rather, we believe that apart from the extra information in the graph, a model that jointly models structure and semantics, like GOFA, is critical.
> > > > >
> > > > > To further justify our statement, we want to highlight Table 4 in the paper, where we compare the LLM-N with second-stage GOFA on downstream graph tasks. The LLM-N is supplied with **exactly the same information** as the input to GOFA, including neighbor node texts, edge connections, and edge texts (similar to what you suggested). In this case, we still observe that GOFA achieves better performance than LLM with much less time consumption. These results can be attributed to 1. The architecture design of GOFA to enable the joint graph structure-language context learning potential; 2. The designed pre-training tasks to empower the GOFA with such abilities; 3. The instruction-tuning stage to allow the model to learn on a similar task format.
> > > > >
> > > > > The GNN-RAG [1] first extracts possible reasoning paths from KG and then feeds it with query into LLM for question answering. Although it also utilizes information from graph, it is not a joint reasoning over both structure and language. For graph tasks that require reasoning over more complex graph structures, like tree or cycle, these types of methods are inherently sub-optimal.
> > > > >
> > > > > 3. **If GOFA is indeed a general-purpose Graph Language Model then it should be able to answer questions about the graph.**
> > > > >
> > > > > Thank you for raising this important point. Based on your suggestion, we have conducted additional experiments on a variety of graph structure tasks, including node, link, and graph-level predictions. To provide a comprehensive comparison, we selected Mistral as the baseline model, and design the prompts for Mistral following the incident prompt format proposed in [2]. Specifically, the graph is described as “[NODE.A] is connected to [NODE.B]”. To ensure a fair comparison, we fine-tuned both Mistral and our GOFA model using the same datasets and tasks, and report the results as below.
> > > > >
> > > > > | Task | Node Degree (Acc) | Shortest Path Distance (Acc) | Node Count (Acc) |
> > > > > | --- | --- | --- | --- |
> > > > > | Task Level | Node | Link | Graph |
> > > > > | Mistral | 98.6 | 99.8 | 99.9 |
> > > > > | GOFA | 98.2 | 99.7 | 86.3 |
> > > > >
> > > > > The results demonstrate that GOFA achieves competitive performance compared to that of LLM baselines in both node-level (node degree) and link-level (shortest path distance) structural tasks, highlighting its strong ability to answer structural questions. We also observe that GOFA is outperformed by LLMs on the graph-level task (node count), and we suspect that because our pre-training tasks focus on understanding local graph structures, GOFA is unfamiliar with outputting global information like node count. In the future, we aim to incorporate more graph-level tasks in our pre-training pipeline to further enhance the model’s capacity to learn graph properties.
> > > > >
> > > > > Reference:
> > > > >
> > > > > [1] Mavromatis, Costas, et al; “GNN-RAG: Graph Neural Retrieval for Large Language Model Reasoning”, ArXiv, 2024.
> > > > >
> > > > > [2] Fatemi, Bahare, et al; “Talk Like a Graph: Encoding Graphs for Large Language Models”, ICLR, 2024.

---

> > > > > > ### Comment · Reviewer_qxFo · 2024-11-27
> > > > > >
> > > > > > Thank you for your responses and your effort! I will update my score. Please include the new evaluations in the final version of the paper too.

---

> > > > > > > ### Author Response · Authors · 2024-11-28
> > > > > > >
> > > > > > > Thanks for your constructive opinions! We will definitely include all the new evaluations in future revisions.

---

> ### Comment · Reviewer_qxFo · 2024-11-24
> **Response to comment 1/3**
>
> Thank you for your response. However, you did not answer my question, and, the evaluation procedure you described above shows that you performed an unfair comparison with the baselines.
>
> Imagine a typical LLM model performing sentence completion, and assume you want to predict a paper's abstract or title. If I just provide a part of the paper's abstract to the LLM, it will probably perform mediocre to the task. Now imagine I, using a citation network, find all related works of that paper and provide also their abstracts/titles to the LLM. For instance, creating a prompt like:
>
> paper 1 - title: ...., abstract: ....., \
> paper 2 - title: ...., abstract: ....., \
> paper 3 - title: ...., abstract: ....., \
> paper 4 - title: ..... abstract:
>
> and letting the LLM fill in the last paper's abstract. Obviously, the results will be better, right? Notice, that the LLM is totally frozen and all I changed was my prompt to include related papers. How is this approach any different than the one you describe in GOFA? Because it appears there is no difference. Hence, my original question: "However, apart from the structural understanding task, it is somewhat unclear how these tasks benefit from graph information. For instance, tasks like sentence completion and question answering might be achievable by directly providing the raw text input to a generic LLM, possibly yielding similar outcomes." which you answered by saying that an LLM that sees only that one paper is performing worse, which is totally expected since it is an unfair comparison as you provide less information to it.
>
> If the whole argument of your work is that extra information is needed (which can be found from another source like a graph), then how is it different from works like this https://arxiv.org/abs/2405.20139 that first extracts all the related information and then provides it to a **frozen** LLM? If GOFA is indeed a general-purpose Graph Language Model then it should be able to answer questions **about** the graph, just like in the case of the shortest path task in Figure 1.

---

### Official Review · Reviewer_sGjN · 2024-11-02

**Soundness:** 3
**Presentation:** 4
**Contribution:** 3
**Rating:** 6
**Confidence:** 4

**Summary:**

This paper extends language modeling to the graph domain and proposes a new generative graph language model, GOFA. The GOFA architecture interleaves frozen LLM layers with trained GNN layers, adding structure-aware processing layers to the LLM while keeping the original LLM intact and providing processing paths for regular language modeling through gating mechanisms. Input graphs are transformed into Text-Attributed Graphs (TAGs), and a next-token-prediction framework is adopted over various graph datasets in natural language, serving as a general self-supervised pretraining objective over graph-structured data. GOFA shows strong performance on benchmark datasets versus previous works building foundation models for graph-structured data using LLMs, and provides a flexible framework for integrating LLMs with GNNs.

**Strengths:**

- The authors offer detailed discussion about the desired attributes of a graph foundation model, including the ability to generalize to new tasks and data domains without additional finetuning of the foundation model, i.e. zero-shot settings.
- The authors also give detailed discussion on previous methods for incorporating GNNs into LLM-based architectures (LLMs as predictors and LLMs as an enhancer), and discuss the drawbacks of previous approaches in building general graph foundation models. The graph foundation model should account for the unique structural information of input graphs and generate graph representations depending on the input prompt. It should preserve the LLM’s prompt following capabilities while also learning graph structure and semantic information jointly.
- The authors propose an architecture which designs a graph language encoder that encodes input TAGs into a fixed number of tokens while accounting for both the semantic information of node and edge text as well as the structure of the graph. Their model controls computational complexity by only doing message-passing on each index of the memory tokens, and feeding a fixed number of vectors to the LLM decoder for the downstream graph task.

**Weaknesses:**

1. It is unclear whether the results in Table 2 of the paper represent a true zero-shot setting, since the authors mention in Section 5.2 and Appendix F.4 that the model is instruction-finetuned on node classification and link prediction examples. The datasets in Table 2 are unseen, however samples of the same task on other datasets has been seen by the model during instruction finetuning, making it seem like this is a generalization task to unseen datasets rather than a zero-shot setting where the model has never seen node classification samples before.
2. Additionally, given that the authors emphasize GOFA as a graph foundation model, more comprehensive benchmarking across more graph datasets would better support this claim. Examples include Amazon Products, molecular graph datasets such as OGBN-molhiv, or protein-protein interaction datasets such as OGBN-proteins. Further benchmarks would better support the claim that GOFA supports diverse downstream tasks on graph-structured data across a variety of graph datasets.
3. The authors do not benchmark against any traditional GNN baselines. The main comparison remains other LLM and LLM-based graph models, however a graph foundation model should compare against traditional GNN baseline models to indicate performance relative to much lower capacity GNN models on the same tasks.

**Questions:**

1. More clarity on the degree to which the GOFA saw examples of node classification and link prediction during instruction finetuning would improve clarity on whether the results can be called a zero-shot setting, or whether GOFA is generalizing seen tasks to new datasets unseen during pretraining and instruction finetuning.
2. More benchmarking results across graph datasets as well as including traditional GNN baselines would greatly strengthen the empirical results of GOFA, and support its claim as a graph foundation model.

---

> ### Author Response · Authors · 2024-11-18
> **Response to Reviewer sGjN (1/2)**
>
> > W1: It is unclear whether the results in Table 2 of the paper represent a true zero-shot setting, since the authors mention in Section 5.2 and Appendix F.4 that the model is instruction-finetuned on node classification and link prediction examples. The datasets in Table 2 are unseen, however samples of the same task on other datasets has been seen by the model during instruction finetuning, making it seem like this is a generalization task to unseen datasets rather than a zero-shot setting where the model has never seen node classification samples before.
> >
> We believe our instruction-tuning and zero-shot evaluation protocol is valid as it exactly follows that of LLMs. An LLM model will also need instruction fine-tuning first to understand the tasks, otherwise it can hardly produce meaningful responses. To be more concrete, LLM is first pre-trained on large corpus to acquire general knowledge of human language. At this stage, the model has minimal question-answering ability, and can hardly generalize to unseen tasks or unseen formats (it is now an autoregressive model that completes sentences only). Next, the model is instruction fine-tuned on a small amount of data with diverse tasks to learn different task formats (these datasets include Alpaca, ColossalChat, etc.). In this step, the purpose is to learn the task format instead of gaining new knowledge [1]. Only after this process, the LLM can be used to answer free-form questions. For example, LLMs are often instruction-tuned on MATH dataset, seeing numerous math problem patterns during training (e.g. determining the number of a type of fruit), yet a new math problem of very similar format is still considered zero-shot learning during evaluation (determining the number of a particular type of colored balls).
>
> In GOFA, this is the same scenario, where the purpose of instruction tuning is to learn the format of the task (e.g. classify paper into categories in a citation network) instead of gaining new knowledge, and we only use a small amount of data compared to pre-training for this purpose. Then, the experiment is conducted on various datasets that the model has never seen before but may share similar task formats to seen datasets (e.g. determine the most likely product type in a product network; both are node classification tasks). Therefore, the evaluation of GOFA is consistent with LLMs and is a true zero-shot setting. This zero-shot setting is also widely accepted by graph foundation model literature such as LlaGA [2], ZeroG [3],  etc.
>
> Moreover, we believe the requirement that a graph foundation model never sees any node classification samples might be overly strict and even infeasible since node classification is the most common type of task for graphs—probably analogous to requiring an LLM to not see any question-answering tasks. We hope this clarifies the rationale behind our methodology and reinforces the validity of our zero-shot evaluation protocol.
>
> >W2: Additionally, given that the authors emphasize GOFA as a graph foundation model, more comprehensive benchmarking across more graph datasets would better support this claim. Examples include Amazon Products, molecular graph datasets such as OGBN-molhiv, or protein-protein interaction datasets such as OGBN-proteins. Further benchmarks would better support the claim that GOFA supports diverse downstream tasks on graph-structured data across a variety of graph datasets.
> >
> Thanks for the suggestion. We have already included Amazon product dataset (Table 2) in our experiment. In general response 1, we additionally conduct zero-shot experiments on protein-protein interaction datasets and molecular graph datasets.
>
> We would like to also point out that, while we are already excited about GOFA’s potential to generalize to unseen domains, a ChatGPT-level graph foundation model still needs to be trained or fine-tuned on much more diverse data in order for the model to generalize to broader domains [4]. The LLM’s generalizability to a particular subject area also comes from the fact that its pre-training process includes data from the proximal area or even the area itself, and it is difficult for LLM to generalize to completely unseen subjects. For example, without including code corpus from open-source platforms, LLMs struggled to solve code-related problems. We believe in the current stage, the proposed graph-language joint modeling problem and the GOFA architecture lay a promising path to a stronger and more generalized model. We will continue improving GOFA with more diverse and comprehensive data.
>
> >W3: The authors do not benchmark against any traditional GNN baselines. The main comparison remains other LLM and LLM-based graph models, however a graph foundation model should compare against traditional GNN baseline models to indicate performance relative to much lower capacity GNN models on the same tasks.
> >
> Thanks for the advice. In general response 2, we add zero-shot GNN baselines for reference.

---

> > ### Author Response · Authors · 2024-11-18
> > **Response to Reviewer sGjN (2/2)**
> >
> > >Q1: More clarity on the degree to which the GOFA saw examples of node classification and link prediction during instruction finetuning would improve clarity on whether the results can be called a zero-shot setting, or whether GOFA is generalizing seen tasks to new datasets unseen during pretraining and instruction fine-tuning.
> > >
> >
> > A: As we discussed in W1, our zero-shot setting is exactly the same as LLMs.
> >
> > >Q2: More benchmarking results across graph datasets as well as including traditional GNN baselines would greatly strengthen the empirical results of GOFA, and support its claim as a graph foundation model.
> > >
> >
> > A: In W2 and W3, we add experiments on molecule and protein datasets and also add traditional GNN baseline for comparison.
> >
> > Reference:
> >
> > [1] Ouyang, Long, et al. “Training language models to follow instructions with human feedback”, ArXiv.
> >
> > [2] Chen, Runjin, et al. “LlaGa: Large Language and Graph Assistant”, ICML24.
> >
> > [3] Li, Yuhan, et al. “ZeroG: Investigating Cross-dataset Zero-shot Transferability in Graphs”, KDD24.
> >
> > [4] Bommasani, Rishi, et al. “On the Opportunities and Risks of Foundation Models”, ArXiv.

---

> ### Author Response · Authors · 2024-11-24
>
> Dear reviewer sGjN,
>
> As the discussion period ends soon, we would like to check whether our responses answer your questions. Following your insightful comments, we have provided a more detailed discussion on the zero-shot capabilities of our model, demonstrating its alignment with the same settings as LLMs. Additionally, we have conducted new experiments on protein-protein and molecule graphs and included more GNN baselines for comparison. Thank you again for your comments and suggestions to improve our paper, and we look forward to your reply.
>
> Best,
>
> Authors

---

> > ### Comment · Reviewer_sGjN · 2024-11-26
> >
> > Thank you for the detailed responses and clarifications.
> >
> > Regarding the instruction-tuning and zero-shot evaluation protocol, I am still not fully convinced that the training and evaluation setup in GOFA can be thought of as zero-shot in the sense of the model performing never-before-seen tasks. I acknowledge the authors' explanation about Large Language Models (LLMs) being instruction finetuned on a small amount of data with diverse tasks in order to train to generate coherent outputs for many tasks. However, it is unusual for the LLM to be instruction finetuned on a significant number of samples of the exact task which it will be evaluated on in downstream evaluations on new datasets, which seems to be the case with node classification and link prediction for GOFA. It is understandable, however, that this is a gray area; more careful phrasing about the setup of the training and evaluation of GOFA would be sufficient to state the performance of GOFA.
> >
> > Overall, the authors presented encouraging new results and addressed some of my concerns, and I will be revising my score to a 6. I would like to see the paper accepted if minor revisions are made to more clearly state to what degree the downstream tasks represent zero-shot evaluations.

---

> > > ### Author Response · Authors · 2024-11-26
> > >
> > > We sincerely appreciate the time and effort you dedicated to reviewing our work, as well as your encouraging and constructive feedback! In response to your valuable suggestions, we have incorporated a discussion on the limitations section, which is highlighted in blue in the updated PDF. Due to the submission deadline, we plan to refine the phrasing in the main text to clarify our definition of "zero-shot" in future revisions.

---

### Official Review · Reviewer_5txS · 2024-11-08

**Soundness:** 3
**Presentation:** 3
**Contribution:** 3
**Rating:** 8
**Confidence:** 2

**Summary:**

The paper proposes a new joint graph and language model that can perform different graph and language tasks. The model is a combination of an LLM compressor, GNN layers and an LLM decoder. The model can be instruction-tuned to solve many graph-structured tasks, and at the same time, it remains effective on language tasks. The model is also more efficient than plain LLMs solving the same tasks, since it separates the input into different nodes and edges.

**Strengths:**

1. The paper addresses an important and challenging problem of building generic and effective models for different modalities.
2. The proposed model, while being a combination of existing methods, is logical and original. Especially, applying the GNN layers on top of memory tokens seems a very good idea to exchange information in an efficient and flexible way.
3. The experiments are sufficiently large, diverse and challenging.
4. The results are overall strong and promising.

**Weaknesses:**

1. The model is designed more for the tasks where nodes/edges are rich in text. It would be interesting and more impactful to include tasks (in the pretraining and/or eval) that are more numeric, e.g. molecule prediction/generation such as ZINC or from open graph benchmark. While the SOTA specialized GNNs are probably going to work better in those cases, it would be interesting to see the gap.

2. Related to above, one potential issue of this approach is that representing node/edge features can be extremely inefficient when the features are high-dimensional (e.g. a node is an image or some other array data).

3. What's the SOTA GNN results in Table 2? It's not a direct comparison, but useful to know for reference.

4. The examples in Fig 2 do not seem to require much knowledge of the graph, i.e. a plain LLM should be able to solve these tasks well. The example does not show what are the nodes and what is the meaning of edges and why those are important for the tasks. It seems like all the information exists within the node.

5. In Table 4, it's unclear if LLM-N and GOFA-F trained on the same data? Also does LLM-N have the complexity of O((V k)^2) as discussed in L298?

Minor:

- In the "LLMs as predictors" some representative papers like GraphLLM are not discussed.

**Questions:**

1. Do the authors use a pretrained llm decoder or only a pretrained compressor?

2. From eq. 3, since GNN is not indexed by k, it appears to be shared across all the K tokens. Is it correct?

3. In the graph task, the authors "set the default target nodes to all nodes in the graph". How the authors combine generations from each node intto a single answer/prediction?

---

> ### Author Response · Authors · 2024-11-18
> **Response to Reviewer 5txS (1/2)**
>
> >W1: The model is designed more for the tasks where nodes/edges are rich in text. It would be interesting and more impactful to include tasks (in the pretraining and/or eval) that are more numeric, e.g. molecule prediction/generation such as ZINC or from open graph benchmark. While the SOTA specialized GNNs are probably going to work better in those cases, it would be interesting to see the gap.
> >
>
> Thanks for bringing this interesting point. In general response 1, we evaluate GOFA on molecule datasets that is more numeric-style. Although the performance is not as good at supervised methods, GOFA still achieve comparable results to other zero-shot baselines.
>
> >W2: Related to above, one potential issue of this approach is that representing node/edge features can be extremely inefficient when the features are high-dimensional (e.g. a node is an image or some other array data).
> >
>
> Thank you for raising this insightful concern. This is indeed a unique challenge when node/edge features are not purely textual-based. However, while our work primarily focuses on graphs within the text or textualized space, which covers most graph-based tasks studied in the research community, we believe that extending this framework to handle general multi-modal graphs is a promising and valuable direction for future research. In particular, GOFA has the potential to serve as a solid foundation for integrating multi-modal models. For instance, high-dimensional node features such as images could be encoded into the text space using vision-language models like BLIP-2 [1] or Flamingo [2], allowing GOFA to seamlessly incorporate these embeddings. Similarly, for array data such as tabular inputs, recent advancements like TableGPT [3] demonstrate the potential to generate robust representations that could be adapted for GOFA. These specialized models can effectively transform diverse modalities into a shared embedding space, enabling GOFA to capture interactions between multi-modal node representations and leverage them for enhanced learning. We appreciate your suggestion and will explore this promising direction in subsequent works.
>
> >W3: What's the SOTA GNN results in Table 2? It's not a direct comparison, but useful to know for reference.
> >
>
> In general response 2, we provide SOTA GNN baselines on zero-shot setting, we also refer you to Appendix B.1 for the supervised experiments as a reference.
>
> >W4: The examples in Fig 2 do not seem to require much knowledge of the graph, i.e. a plain LLM should be able to solve these tasks well. The example does not show what are the nodes and what is the meaning of edges and why those are important for the tasks. It seems like all the information exists within the node.
> >
>
> Thanks for the suggestion. The example shown in Fig 2 is mainly for demonstrating the functionality of Node-Of-Generation (NOG). We have updated Fig 2 to include a task that better utilizes graph structure for sentence completion (see the revised PDF, page 4, with changes marked in blue).
>
> Additionally, we have included further examples from real-world graphs in Fig 7 (Appendix, page 19). For instance, in the product graph example, nodes represent different products and their descriptions, while edges indicate co-purchase relationships. By connecting an NOG and asking GOFA to find the majority category in the graph, the GOFA can correctly answer the corresponding category. In this example, without information from all nodes in the graph, it is hard for the model to generate the correct answer.
>
> >W5: In Table 4, it's unclear if LLM-N and GOFA-F trained on the same data? Also does LLM-N have the complexity of O((V k)^2) as discussed in L298?
> >
>
> In Table 4, the input to LLM-N and GOFA contains exactly the same amount of information and this experiment is to show that GOFA’s power does not solely come from the extra neighborhood information but also comes from its strong structural modeling ability. The LLM here is not fine-tuned. The main reason we didn’t tune the LLM is that if we input the same amount of information, the tuning of the LLM will result in OOM even if we use a very small $r$ in LoRA. This again demonstrates the memory efficiency of the GOFA compared to the original LLM.
>
> And you are correct that LLM-N has the complexity of $O((V k)^2)$, where $V$ is the number of nodes and $k$ is the average number of tokens per node. This complexity arises from the self-attention mechanism, which scales quadratically with the input sizes.

---

> > ### Author Response · Authors · 2024-11-18
> > **Response to Reviewer 5txS (2/2)**
> >
> > >Q1: Do the authors use a pretrained llm decoder or only a pretrained compressor?
> > >
> >
> > A: Yes, we use the pre-trained LLM decoder from Mistral-v0.2 and pre-trained LLM encoder from ICAE.
> >
> > >Q2: From eq. 3, since GNN is not indexed by k, it appears to be shared across all the K tokens. Is it correct?
> > >
> >
> > A: Yes, the weight of GNN layers are shared across different token indexes.
> >
> > >Q3: In the graph task, the authors "set the default target nodes to all nodes in the graph". How the authors combine generations from each node into a single answer/prediction?
> > >
> >
> > A: For all tasks, we introduced a prompt node whose feature is the task query to serve as the NOG (discussed in the Section 3.3). For graph tasks, the prompt node will connect to all nodes in the graphs, such that the message will be exchanged between the prompt node and all nodes in the graph. Finally, we use the output representation of the prompt node as the input to the decoder.
> >
> > Reference:
> >
> > [1] Li, Junnan, et al. “BLIP-2: Bootstrapping Language-Image Pre-training with Frozen Image Encoders and Large Language Models”, ICML23.
> >
> > [2] Alayrac, J., et al. “Flamingo: a Visual Language Model for Few-Shot Learning”. NeurIPS22.
> >
> > [3] Zha, Liangyu, et al. “TableGPT: Towards Unifying Tables, Nature Language and Commands into One GPT”, ArXiv.

---

> ### Author Response · Authors · 2024-11-24
>
> Dear reviewer 5txS,
>
> We sincerely appreciate your positive opinions and constructive review of our paper. As the discussion period is near its end, we would like to ensure our response aligns with your expectations and addresses your concerns. In our response, we conduct additional experiments on molecule datasets and add baselines for zero-shot GNN methods. We further clarified the importance of graph context in the graph task and explained the scenarios of how GOFA can be applied to much broader multi-modalities. We appreciate your feedback and look forward to any further comments.
>
> Best,
>
> Authors

---

### Author Response · Authors · 2024-11-18
**General Response to All Reviewers**

We would like to express our sincere gratitude to all reviewers for the valuable time you spent evaluating our paper. We are particularly grateful for your recognition of our contribution:

- Our work identifies “three crucial properties of an effective GFM” (Reviewer 5YVL) and offers a “detailed discussion” on how it is critical for generalization ability and “zero-shot setting” (Reviewer sGjN). We also thank Reviewer 5YVL for agreeing with us that these properties are “important considerations”.
- The proposed GOFA is “logical and original” (Reviewer 5txS). It is a “very good idea” to alternate GNN and LLM layers and apply GNN on top of the memory token (Reviewer 5txS). GOFA “controls the complexity by only doing message-passing on each memory token index and fed fixed number of tokens to LLM decoder”, which is both “efficient and flexible” for modeling text-attributed graphs (Reviewer 5txS, sGjN).
- The experiments are “sufficiently large, diverse and challenging” (Reviewer 5txS, qxFo). GOFA achieves “impressive and strong results” across different experiments (Reviewer 5txS, 5YVL). Particularly, the experiments validate that the training of GOFA “does not degrade the performance of the original pre-trained LLMs” (Reviewer 5YVL). Overall, the results support that GOFA has great potential to be a “general-purpose Large Graph Language Model” (Reviewer qxFo).
- We also thank you for describing our paper as “well-structured”, and “easy to follow” (Reviewer 5YVL, qxFo)

You also expressed some concerns about our work. We would like to address a few common concerns here and leave others in individual responses.

**R1: Additional experiment on molecule and protein datasets.**

According to the reviewers’ suggestions, we add experiments on molecule datasets. We include evaluations on ogbg-molhiv and BBBP datasets. We closely follow the setting of GIMLET [1]. Specifically, we first instruction fine-tune the pre-trained GOFA on randomly sampled 100,000 question-answering pairs from the Chembl-pretrain dataset. The question concentrated on asking about different molecule properties. Next, we evaluate the fine-tuned GOFA on ogbg-molhiv and BBBP in **zero-shot** setting. We use AUROC as the evaluation metric. Here is the result:

| AUROC | BBBP | HIV |
| --- | --- | --- |
| OFA[5] | - | 35.67 |
| MoMu[6] | 49.81 | 50.26 |
| Galactica[7] | 53.94 | 33.85 |
| GIMLET[1] | 59.39 | 66.24 |
| GOFA | 54.91 | 53.02 |

We directly report results of OFA [5] in their paper and all other baselines are referred from GIMLET [1]. We can see that with only 100,000 instruction-tuning samples, GOFA already achieves comparable results to baselines. Note that GIMLET fine-tunes their model on the whole Chembl-pretrain dataset, which contains more than 400 million question-answering pairs.

For protein dataset, ogbn-PPI dataset does not contain textual features (the information that relates a node id to an actual protein is missing). Therefore, we manually collect a human protein-protein interaction dataset from [2]. The dataset contains 36,630 interactions between different proteins. The task aims to predict the interaction between proteins. Identical to cora-link, we randomly split interactions into train/val/test set with a ratio of 0.85/0.05/0.1. Next, we directly evaluate the **zero-shot** performance of both GOFA and LLMs on the test set. We use ROC-AUC as the evaluation metric:

| Protein | Model | AUROC |
| --- | --- | --- |
| Zero-shot | LLM-N | 35.24 |
| Zero-shot | GOFA-T | 64.59 |

We can see without any further adaption, GOFA already achieves good performance in a zero-shot way, outperforming LLM alone by a large margin.

**R2: More GNN baselines.**

We include more GNN baselines under the zero-shot setting. Since models like GCN and GAT cannot do zero-shot tasks, we compare against graph self-supervised learning baselines. These methods learn node embeddings, which are compared to label text embeddings to make predictions based on similarity. Specifically, we include DGI [3] and GraphMAE2 [4] and follow the same evaluation procedure. The results are summarized below:

|  | Cora-Node (7 ways) | WikiCS (10 ways) | Products (10 ways) |
| --- | --- | --- | --- |
| DGI[3] | 20.15 | 13.63 | 21.35 |
| GraphMAE2[4] | 37.08 | 19.30 | 25.46 |
| Best LLM baselines | 60.54 | 63.63 | 70.16 |
| Best Graph-LLM baselines | 69.53 | 43.45 | 66.07 |
| GOFA-T | 70.81 | 71.17 | 79.33 |

As shown in the results, GOFA significantly outperforms methods that use GNNs for self-supervised learning in the zero-shot setting. It also surpasses LLM and Graph-LLM baselines (as presented in Table 2), demonstrating its strong zero-shot generalization ability.

For the supervised traditional GNN performance, we kindly refer reviewers to Appendix B.1.

---

> ### Author Response · Authors · 2024-11-18
>
> Reference:
>
> [1] Zhao, Haiteng, et al. “GIMLET: A Unified Graph-Text Model for Instruction-Based Molecule Zero-Shot Learning”, ICLR24.
>
> [2] Pan, Xiao-Yong, et al. “Large-scale prediction of human protein-protein interactions from amino acid sequence based on latent topic features”, Journal of Proteome Research.
>
> [3] Veličković, Petar, et al. “Deep Graph Informax”, ICLR19.
>
> [4] Hou, Zhenyu, et al. “GraphMAE2: A Decoding-Enhanced Masked Self-Supervised Graph Learner”, WWW23.
>
> [5] Liu, Hao, et al. “One For All: Towards Training One Graph Model for All Classification Tasks”, ICLR24.
>
> [6] Su, Bing, et al. “A Molecular Multimodal Foundation Model Associating Molecule Graphs with Natural Language”, Arxiv22.
>
> [7] Taylor, Ross, et al. “Galactica: A large language model for science.”, Arxiv22.

---

### Meta-Review · Area_Chair_h4R2 · 2024-12-22

**Metareview:**

This paper presents GOFA, a generative model for joint graph and language modeling that interleaves GNN layers with a frozen LLM to address graph-structured tasks effectively. Reviewers commended the paper for its relevance, strong experimental results, and well-structured presentation. However, concerns were raised about the clarity of the zero-shot evaluation protocol, the necessity of specific pretraining tasks, and benchmarking against baselines. The authors addressed these issues with extensive responses, additional experiments, and clarifications, which resolved some but not all doubts. The final recommendation leans towards acceptance as a poster presentation, respecting the positive consensus.

**Additional Comments On Reviewer Discussion:**

During the discussion phase, reviewers highlighted critical points, including the ambiguity surrounding the zero-shot evaluation setup, the importance of graph-specific tasks, and comparisons with related methods like GraphLLM. The authors clarified their approach by situating the zero-shot protocol in the context of established LLM benchmarks and emphasizing the role of pretraining tasks in enhancing generalization. They also included new experiments comparing GOFA against traditional GNNs and updated the manuscript to address reviewer-specific questions about its structural design and benchmarks. While these efforts improved the paper's clarity and addressed some concerns, remaining questions about its broader applicability and evaluation left minor reservations. Nonetheless, the reviewers' overall positive assessments guided the decision.

---

### Decision · Program_Chairs · 2025-01-22

Accept (Poster)